# Subsurface geological and geophysical data from the Po Plain and the northern Adriatic Sea (north Italy)

Michele Livani [1], Lorenzo Petracchini[1], Christoforos Benetatos[2], Francesco Marzano[2], Andrea Billi[1], Eugenio Carminati[3], Carlo Doglioni[3,4], Patrizio Petricca[5], Roberta Maffucci[3,4], Giulia Codegone[2], Vera Rocca[2], Francesca Verga[2], Ilaria Antoncecchi[6]

[1]Consiglio Nazionale delle Ricerche, IGAG, Roma, Italy
[2]Politecnico di Torino, Dipartimento di Ingegneria dell'Ambiente, del Territorio e delle Infrastrutture, Torino, Italy
[3]Sapienza Università di Roma, Dipartimento di Scienze della Terra, Roma, Italy
[4]Istituto Nazionale di Geofisica e Vulcanologia, Osservatorio Nazionale Terremoti, Roma, Italy
[5]ISPRA Servizio Geologico d'Italia, Via Vitaliano Brancati, 48, 00144 Roma, Italy
[6]Ministero della Transizione Ecologica, Direzione Generale per le Infrastrutture e la Sicurezza, Roma, Italy

*Correspondence to*: Lorenzo Petracchini (lorenzo.petracchini@igag.cnr.it)

**Abstract.** The Po Plain (Italy) is one of the most densely populated and productive regions of Europe, characterized by a flourishing economy (also linked to strategic subsurface resources) and several world cultural and natural heritage sites. The coupling of social-economic interests with geological hazards (i.e., seismic, subsidence and flooding hazards) in this area requires accurate knowledge of the subsurface geology, active geological processes, and impact of human activities on natural environments to mitigate the potential natural and anthropic risks.

Most data unveiling the subsurface geology of this region were produced by the hydrocarbon exploration industry. Po Plain hosts indeed many hydrocarbon fields that were discovered since the early 1950s giving rise to the subsurface exploration through extensive seismic reflection surveys and drilling of numerous deep wells. In this work, geological-geophysical data from 160 deep wells drilled for hydrocarbon exploration/exploitation purposes in the Po Plain and in the facing northern Adriatic Sea have been collected and digitized along with several published geological cross-sections and maps. These data have been used to reconstruct the overall subsurface 3D architecture and to extract the physical properties of the subsurface geological units.

The digitized data are suitable to be imported into geo-software environments so to derive the geophysical-mechanical properties of the geological units for a wealth of applied and scientific studies such as geomechanical, geophysical and seismological studies.

The integrated dataset may represent a useful tool in defining regional first order strategies to ensure the safety of the urbanized areas and human activities and to reduce natural and anthropic risks that may affect this crucial region of Europe. In particular, the data collected would be useful to highlight sensible areas where data collection and more detailed studies are needed. Nowadays, such issues are particularly relevant for the underground industry development related to the increasing interest on possible $CO_2$ and hydrogen underground storage, which can play a fundamental role in the energy transition process towards the decarbonisation goals.

# 1 Introduction

The Po Plain in the north of Italy is the most productive and prosperous region of Italy (Fig. 1a), with a per capita income similar to that of central and northern European countries (https://ec.europa.eu/eurostat; Helliwell and Putnam, 1995; Tabellini, 2010; European Commission, 2016). The area is densely populated (https://ec.europa.eu/eurostat; Fig. 1a) and hosts numerous UNESCO world heritage sites such as the cities of Venice, Verona, Bergamo, and Ravenna (https://whc.unesco.org/; Fig. 1b). Since the second half of the 20th century, the discovery and exploitation of numerous hydrocarbon (mostly gas) resources

contributed to the economic development of Italy. The Po Plain and the facing northern Adriatic Sea host indeed the majority of hydrocarbon fields in the country (extracting almost 33% of the total national gas production) and most of the Italian underground gas storage sites, which have been operative since the 1960s (https://unmig.mise.gov.it/; Fig. 1b).

The Eni-Agip Company hydrocarbon exploration and exploitation activities in the Po Plain and northern Adriatic Sea led to the production of a large amount of subsurface data, which includes: (i) seismic data acquired through extensive regional 2D

and 3D seismic surveys for the development of onshore/offshore hydrocarbon fields and (ii) well data acquired during the drilling of explorative and development wells. The subsurface dataset provided structural, stratigraphic, and sedimentological information, which allowed the accurate knowledge of the regional subsurface architecture and geological evolution (e.g., Pieri and Groppi, 1981; Cassano et al., 1986; Casero, 2004; Ghielmi et al., 2010, 2013; Fantoni et al., 2009, 2010).

The geological evolution of the Po Plain is accompanied by seismic, subsidence and flooding events that are the main

phenomena to consider in natural hazards assessment. The cultural, social, and economic relevance of this region asks for a careful evaluation of both natural hazards and impact of human activities to mitigate the potential correlated risks and guarantee the safety of the urbanized areas and human activities themselves.

The Po Plain is characterized by moderate seismicity (International Commission on Hydrocarbon Exploration and Seismicity in the Emilia region, 2014). Nonetheless, instrumental and historical intermediate-strong (Mw≥5.0) earthquakes hit the area

(Rovida et al., 2022), such as the events occurred in 1117 (Mw 6.5; Verona area), in 1570 (Mw 5.6; Ferrara area), and 2012 when a seismic sequence in the Modena and Ferrara provinces occurred (main shock Mw 6.09). The latter event promoted the implementation of the seismic monitoring network as an essential instrument for the safe management of industrial activities. Notice that no case of anthropogenically-induced seismicity has been documented so far in the study area (Braun et al., 2020). The Po Plain is also characterized by significant subsidence of both natural and anthropic origins, particularly intense after the

economic growth following the World War II. The natural component is the result of several geological processes, such as sediment load and compaction, vertical tectonic movements, and rebound effects after the last deglaciation (Carminati and Di Donato, 1999). Anthropogenic subsidence is primarily influenced by groundwater withdrawal for industrial, agricultural, and civil uses (e.g., Herrera-Garcia et al., 2021). Subordinately, it is linked to hydrocarbon (mainly gas) exploitation from onshore and offshore reservoirs (Carminati and Martinelli, 2002; Bitelli et al., 2020).

Subsidence mainly affects the central and eastern sectors of the Po Plain: the highest total subsidence rates, greater than -60 mm/yr, were evaluated for the Bologna city area (Carminati and Martinelli, 2002; Zerbini et al., 2007; Baldi et al., 2009),

whereas the highest natural component reaches -2.0 mm/yr at the pede-Apennine zone, near the city of Bologna. The difference between the total (natural plus anthropogenic) vs natural components of the present-day subsidence shows that the main factor controlling modern subsidence in this region is anthropogenic (Carminati and Martinelli, 2002).

Subsidence is monitored through the implementation of different technologies (GPS, InSAR, levelling surveys) useful to prevent, mitigate, and control the natural processes as well as the human ones (e.g., Dacome et al., 2015; Benetatos et al., 2020).

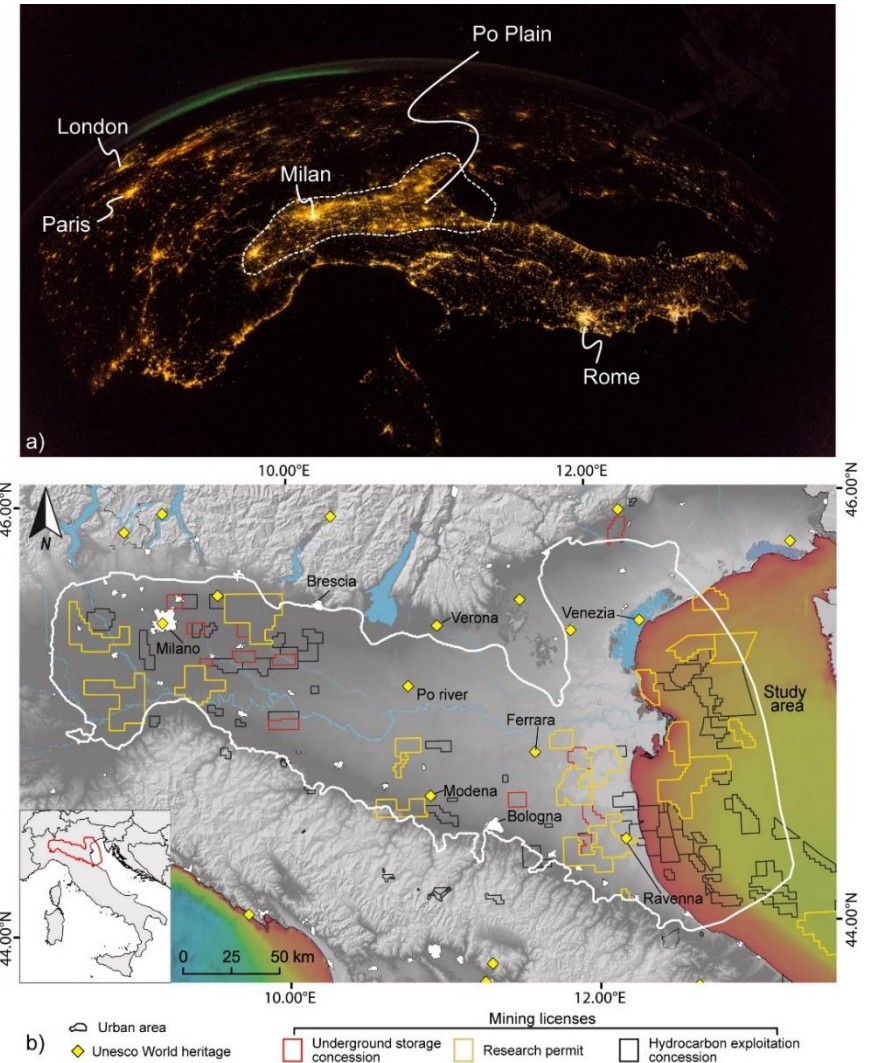

**Figure 1: (a) Modified photo of the ESA astronaut Alexander Gerst, snapped from the International Space Station, that clearly**
**reveals the high-density population in the Po Plain area highlighted by the strong night-light intensity (credit: ESA/NASA - CC BY-SA IGO 3.0; the original photo has been modified). (b) The figure shows the main cities in the Po Plain, the UNESCO world heritages (data from https://whc.unesco.org/), the current mining licenses from https://unmig.mise.gov.it/.**

Flood events related to the Po River (which crosses the whole Po Plain) and its tributaries represent an additional natural risk (Domeneghetti et al., 2015 and references therein). In the past century, the worldwide increase of flood susceptibility and risk increased together with the rise of subsidence due to groundwater depletion (e.g., Herrera-Garcia et al., 2021). In the Po Plain, subsidence and flood hazards are presumably linked since there is a clear-cut correlation between high flood frequency and rapid subsidence, whereas only a few floods occurred in low subsidence areas (Carminati and Martinelli, 2002). Nevertheless, in the first half of 2022, the most significant drought since at least 70 years was observed for the area, causing extensive damage to agriculture, and encouraging the entry of the saline wedge at the mouth of the Po River along the Adriatic coast.

The protection of the cultural, social, and economic heritage of the Po Plain makes the adoption of all available measures necessary to reduce the impact of natural and anthropic-derived hazards in the study area.

To this end, the definition of the physical and geometrical attributes of the outcropping and subsurface geological units provides fundamental knowledge for several scientific issues and could be used in preliminary definition of engineering operations. As an example, the integration of physical parameters of subsurface geological units into a well-defined 3D model can be applied to (i) reduce the uncertainties for earthquake location, contributing to the calculation of more accurate focal mechanisms and performing wave-propagation and ground-motion simulations (e.g., Magistrale et al., 1996; Süss et al., 2001; Molinari et al., 2015; Livani et al., 2022), and (ii) understand, simulate, and predict the response of the geological body to subsurface natural and anthropic processes. The latter is the case of 3D geomechanical numerical models, which represent effective tools to evaluate and predict the possible effects - both at the surface and in the subsurface - of geofluid extraction and storage, to guarantee a safe management of such activities as well as to quantify and better understand the ongoing geological processes (e.g., tectonic deformation, natural subsidence, etc.; Teatini et al., 2006; Codegone et al., 2016; Benetatos et al., 2020). As an example, numerical models can play a fundamental role during the geological sequestration of $CO_2$ and gas storage (e.g., hydrogen as an energy carrier) in natural underground formations since they are mandatory for the optimization of the development strategies, the maximization of storage efficiency, and monitoring activities.

In this paper, we present an integrated database of geological-geophysical data regarding the subsurface of the Po Plain and the facing Adriatic Sea. The database provides a collection of data distinguished into primitive and derived. The primitive data consist of a detailed and accurate collection and digitization of subsurface information extracted from wells, geological cross-sections, and geological maps. The derived data are obtained from the revision and processing of the primitive ones and by gridded surfaces representing the main geological units of the Po Plain subsurface.

It is worth mentioning that, in the study area, the geological literature offering interpretations of the subsoil is abundant in terms of geological reconstructions. Our database represents a collection of the main published works regarding the Po Plain.Detailed studies related to specific sectors of the Po Plain might not be present in our database. Recently, other subsurface geological models have been elaborated providing several isobath maps of the Po Plain (e.g., Turrini et al., 2014; ISPRA, 2015; Molinari et al., 2015; Amadori, et al., 2019, D'Ambrogi et al., 2023). It is worth mentioning two projects: the recent GO-PEG project (D'Ambrogi et al., 2023) ), which provides the geometry of four stratigraphic horizons in the Po Plain area deriving from GeoMol project (http://www.geomol.eu; ISPRA, 2015; Maesano and D'Ambrogi, 2016) and GeoERA-HotLime

project (https://geoera.eu/projects/hotlime6/),and the Mambo project (Molinari et al., 2015), which covers almost the entire Po Plain. In our work, due to technical difficulties (e.g., the impossibility to precisely define the depth reported in the maps), we were not able to integrate all the previous datasets. Anyhow, where possible, we integrated the available regional geological models with other primitive public data to coherently define the geometry of five surfaces of the Po Plain subsurface, which are from the oldest to the youngest: the magnetic basement top, the top of the carbonate succession, the Pliocene base, the Calabrian base, and the base of recent continental deposits. The above mentioned surfaces represent lithological boundaries rather than chronostratigraphic/formational limits, that define units that are expected to show different mechanical behaviour. Most importantly, the gridded surfaces of our dataset come with a series of detailed geophysical and geological parameters extracted from the composite logs of deep wells (i.e., the primitive data). In addition to the geophysical and geological data, a statistical analysis is reported and discussed as a preliminary elaboration of the collected data.

We believe that our dataset provides an important contribution to a broad audience of policymakers and scientists to understand and evaluate geological and anthropic processes in the area, and to set secure developing strategies for correct territory management and to reduce social and economic risks in a strategic area for Italy and Europe.

## 2 Geological setting

The Po Plain and the facing Adriatic Sea lie on the buried sector of the Adria microplate, a promontory of the Africa plate or an independent microplate, interposed between the NE-verging northern Apennines and the S-verging Southern Alps (Fig. 2; Dercourt et al., 1986; Pieri and Groppi, 1981; Castellarin et al., 1985, 1986; Doglioni, 1993; Carminati et al., 2003; Carminati and Doglioni, 2012; Fantoni and Franciosi, 2009, 2010; Turrini et al., 2016; Pezzo et al., 2020). The development of these two facing fold-and-thrust belts, connected with the broad collision of the Eurasian and African plates, led to the formation of the Po Plain basin representing the foreland/foredeep basin of both orogens. Compressional tectonics affected the area since middle Eocene time, with the development of WNW-ESE oriented thrusts in the Southern Alps followed, from Oligocene-lower Miocene onward, by the NW-SE oriented thrust system of the northern Apennines (Coward et al., 1989; Carminati and Doglioni, 2012; Carminati et al., 2012; Maesano and D'Ambrogi, 2016).

The structural and sedimentary framework of the area has been constrained using numerous seismic reflection profiles and deep well logs (e.g., Pieri, 1983; Cassano et al., 1986; Fantoni and Franciosi, 2010; Turrini et al., 2014; ISPRA, 2015; Livani et al., 2018; Amadori et al., 2019, D'Ambrogi et al., 2023 and reference therein).

The Apennines front is characterized by three main orogenic arcs, from West to East: Monferrato, Emilia, and Ferrara Arcs (Fig. 2a). The Southern Alps represent the non-metamorphic retrobelt of the double-verging Alpine chain and, on the western side of the study area, it reaches the southernmost extent with its edge very close to the northern Apennines front (Ravaglia et al., 2006; Fantoni and Franciosi, 2010; Toscani et al., 2014; Fig. 2b). The external fronts of the two facing chains are mostly buried under a siliciclastic sequence (late Eocene-actual) consisting of syntectonic sediments and recent alluvial sediments of the Po River (Pieri and Groppi,1981; Boccaletti et al., 1985; Bigi et al., 1990; Fantoni and Franciosi, 2010; Ghielmi et al., 2010, 2013; Carminati and Doglioni, 2012; Amadori et al., 2019 and reference therein). In detail, the siliciclastic sequence

(Fig. 3) can be subdivided into a lower (late Eocene-early Messinian) and an upper (late Messinian to present) cycle (Ricci Lucchi, 1986). The lower cycle, primarily fed by the Alpine chain, consists of silty and shaly deposits (i.e., Gallare Marls; late Eocene-to-Miocene time), in places intercalated by or interdigitated with sandy and conglomeratic deposits (i.e., Gonfolite fm.; Oligocene time), passing upward to sandy marls (Marnoso-Arenacea fm.; Langhian-to-Messinian time), clays (i.e., Colombacci fm.; Messinian time), and evaporitic deposits (i.e., Gessoso Solfifera fm.; Messinian time). The upper cycle, fed

by both the northern Apennines and the southern Alps, mainly consists of marine sandy and conglomeratic formations (e.g., Sergnano Gravel, Porto Corsini, Porto Garibaldi, Santerno and Asti Sandstones; Pliocene-to-middle/late Pleistocene time) and alluvial deposits (middle/late Pleistocene-to-present; e.g., Muttoni et al., 2003; Garzanti et al., 2011; Ghielmi et al., 2010, 2013; Livani et al., 2018). South of the Po River, the continental deposits consist of alluvial fan and plain deposits embedded in clays and showing elongated shapes, whereas, to the north of the Po River, the sedimentary bodies are wider, generally

tabular and with minor amounts of fine-grained sediments (Ori, 1993; Amorosi and Milli, 2001). These clastic sequences are superimposed on a carbonate and marly substratum (Triassic-middle Eocene), which lies on top of the platform and continental Permian-Triassic formations, lying in turn on the Variscan crystalline basement (for more details about the carbonate sequence refers to Livani et al., 2018 and references therein). The Triassic deposits are sometimes interposed by intra-sedimentary volcanic bodies (e.g., Pieri and Groppi, 1981; Castellarin, 1985; Cassano et al., 1986; Ghielmi et al., 2010; Livani et al., 2018).

The Southern Alps buried front consists of a repetition of Cenozoic clastic units stacked above a regional detachment in the Marne di Gallare fm. (late Eocene) and followed at depth by deep thrust cutting the Mesozoic carbonates and the Variscan crystalline basement (Fantoni et al., 2004; Figs. 2b and 3).

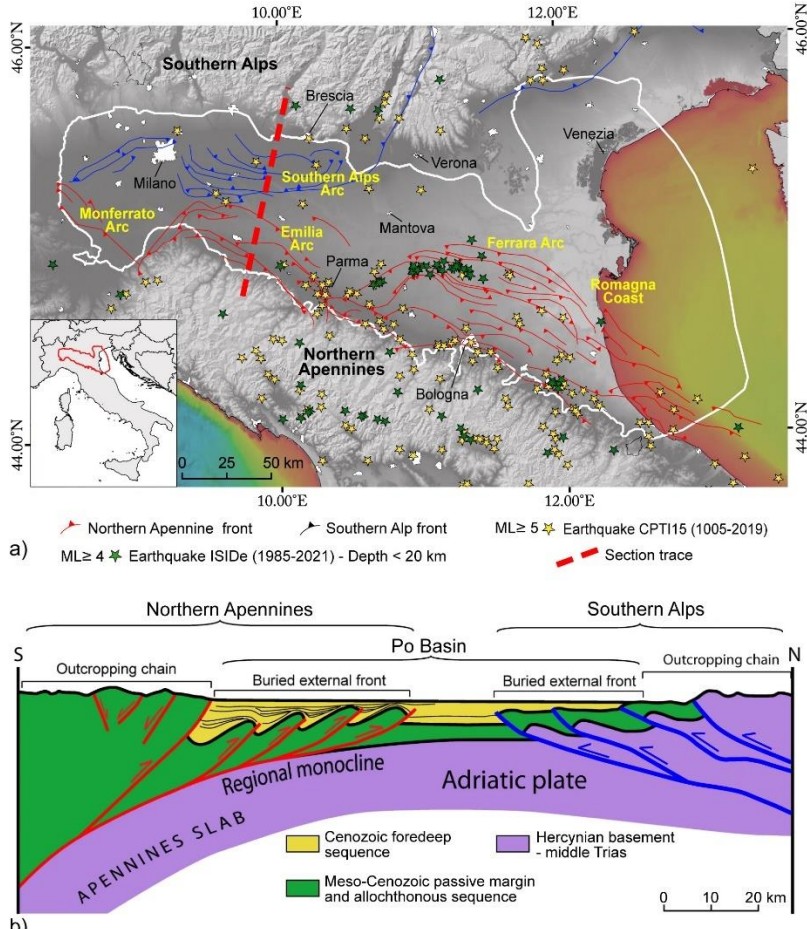

**Figure 2: Study area and structural geology of the Po Plain. (a) Simplified structural map of the Po Plain (modified after Livani et al., 2018). The main buried thrusts of the northern Apennines (red colour) and Southern Alps (blue colour) fronts are shown along with the instrumental and historical seismicity (ISIDe Working Group - INGV, 2010; Rovida et al., 2022). The white polygon represents the study area. (b) Schematic geological cross-section across the Po Plain along a N-S oriented line (by Livani et al., 2018). The Northern Apennines (on the left) and Southern Alps (on the right) fronts can be identified under the Plio-Pleistocene sedimentary cover (in yellow) filling the Po Plain foredeep. Section trace is reported with a red dashed line in Fig. 2a.**

The northern Apennine chain developed by off-scraping the Meso-Cenozoic sedimentary cover of the subducting Adria plate made up of siliciclastic, evaporitic, and shallow to deep-water carbonate deposits (Cati et al., 1987; Bertotti et al., 1993; Casero et al., 1990; Zappaterra, 1990; Grandić et al., 2002; Fantoni and Scotti, 2003; Fantoni and Franciosi, 2010; Masetti et al., 2012; Fig. 3).

Due to the Apennine compressional deformation, the northern Apennines thrust system migrated toward the foreland over time (among many others, Malinverno and Ryan, 1986; Doglioni, 1991; Patacca et al, 1990; Royden et al., 1988; Faccenna et al., 2003; Rosenbaum and Lister, 2004; Scrocca et al., 2006, 2007; Carminati and Doglioni, 2012 and references therein). The convergence is still active as indicated by the moderate-to-high seismicity which historically affects the Po Plain (maximum Mw between 5.5 and 6.5; Rovida et al., 2020, 2022). Studies on the active stress field in Italy (Montone et al., 2004; Devoti et

al., 2008; Cuffaro et al., 2010; Montone et al., 2012), based on the analysis of earthquake focal mechanisms, GPS records, and

borehole breakout data, identify active compression in the shallow portion of the Po Plain with the maximum shortening axis

orthogonal to the main orogenic structures (i.e., NNE-SSW). Locally, the young land morphologies, the presence of different

orders of fluvial terraces, and the deviation of some rivers (including the Po River) near the buried active tectonic structures

are the tangible evidence of the recent tectonic activity (e.g., Burrato et al., 1999, 2003; Boccaletti et al., 2004a, 2004b; Wilson

et al., 2009, Livio et al., 2009; Zuffetti and Bersezio, 2020; Bresciani and Perotti, 2014).

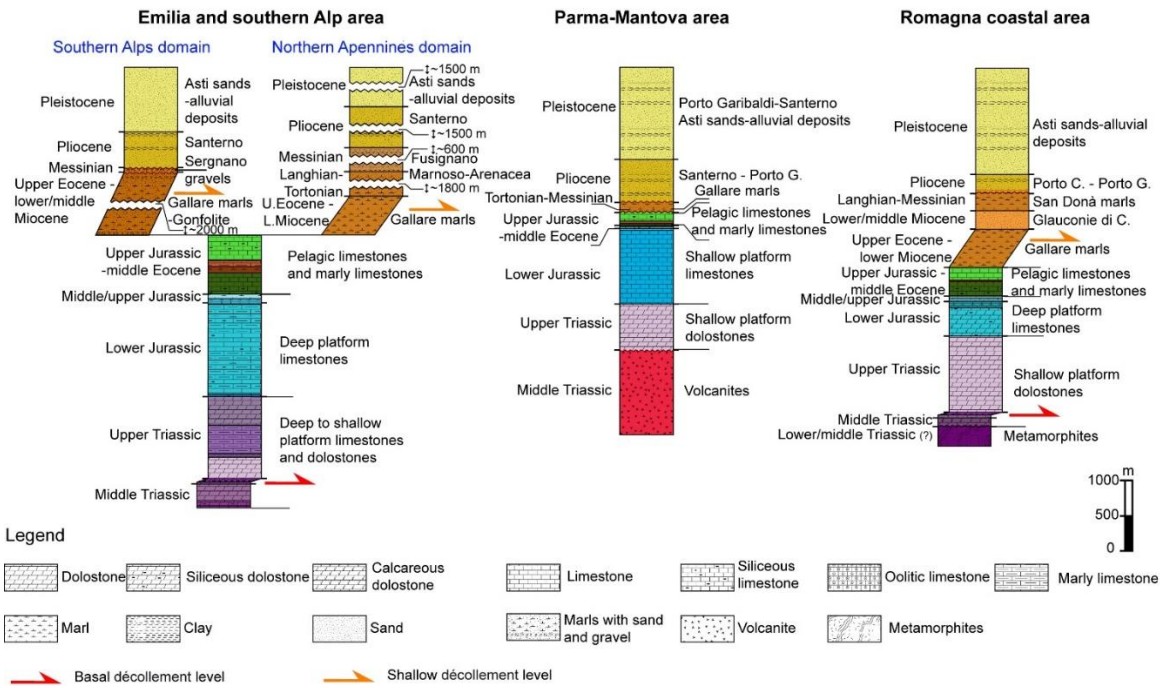

**Figure 3: Synthetic stratigraphic columns of the Po Plain. From left to right: the stratigraphy of Emilia and Southern Alps, Parma-Mantova, and Romagna coastal areas is shown (see Fig. 2a). The carbonate passive margin (from violet to green colours) and siliciclastic active margin (from brown to light yellow colours) sequences are distinguished. The main décollement levels are highlighted (red for basal décollement and orange for shallow décollement). The stratigraphic column to the left shows the different**

**filling and the different formation thickness of the Southern Alps and the northern Apennines foredeep.**

### 3 Database description

We realized our database by collecting, revising, and digitizing geological and geophysical data, originally in raster format,

derived from public sources (i.e., databases and literature works). We collected 160 deep well composite logs, 61 published

geological cross-sections, and 10 geological maps (Fig. 4; Tables 1-3). The data were georeferenced to a common geographical

system (WGS 84/UTM zone 32N; EPSG: 32632) by using the Open Source QGis software (version 3.12.3;

http://www.qgis.org) and then uploaded into a 3D geological modelling software (Petrel® Software, version 2016.2). We

organized the database into two groups, namely "primitive data" and "derived data", containing further hierarchical

subdivisions created for a better organization and comprehension of the database.

The primitive data contain the product of the digitization of the collected isobath maps, the geological cross-sections, and the well data realized and acquired over a long period and for different purposes. The well data include specific sets of well logs aimed to geological/mechanical characterization of the geological units. Therefore, to achieve the integration of the different sets of data, we analysed them focusing on the stratigraphy and geological age of the interpreted units. Notice that in 2009, the Executive Committee of the International Union of Geological Sciences (IUGS) ratified a new subdivision of the Quaternary Period and the Pleistocene Epoch lowering the age of their base from the top of the Gelasian (1.8 Ma) to its base (2.58 Ma; Gibbard et al., 2010). Most of the collected data (Fig. 4; Table 1; Table 2) refer to the pre-2009 chronostratigraphic subdivisionincluding the Gelasian age in the upper Pliocene (Rio et al., 1998). On the contrary, some recent works reinterpreted the sedimentary sequence in the Po Plain and the nearby northern Adriatic Sea (e.g., Ghielmi et al., 2010, 2013; ISPRA, 2015; Maesano and D'Ambrogi, 2016; Amadori et al., 2019) using the new chronostratigraphic subdivision. In our database, since the collected primitive data (Fig. 4; Table 1; Table 2) are both prior and successive to the 2009 chronostratigraphic subdivision, we homogenized the data considering the pre-2009 Pleistocene base as Calabrian base.

We performed an accuracy analysis on the primitive data that unravelled several discrepancies in the interpretation of the subsurface geological horizons. Starting from these observations and considering the well data as the best constraint, we filtered the primitive isobath maps and the geological cross-sections to obtain a coherent dataset. The derived data consist of these filtered isobath maps and the geological cross-sections plus a series of regional surfaces of the main geological units of the Po Plain subsurface. These surfaces were generated without considering the faults occurrence/displacements and they were gridded by means of interpolation of filtered primitive data.

All data (both primitive and derived) are provided in delimited text file format organized according to the data type (i.e., well, geological cross-section, map or gridded surface).

During the processes of data collection and revision, some published subsurface data reconstruction might have not been included in the database for technical reasons such as isobath maps published at an inappropriate resolution, depth contour lines not properly stated, or geological cross-sections with a different lithostratigraphic and structural scheme compared to the one used in this work.

Figure 5 summarizes the technical procedures and the workflow used to process the data and to develop the database. In the following paragraphs, the methods and the produced primitive data are explained in detail.

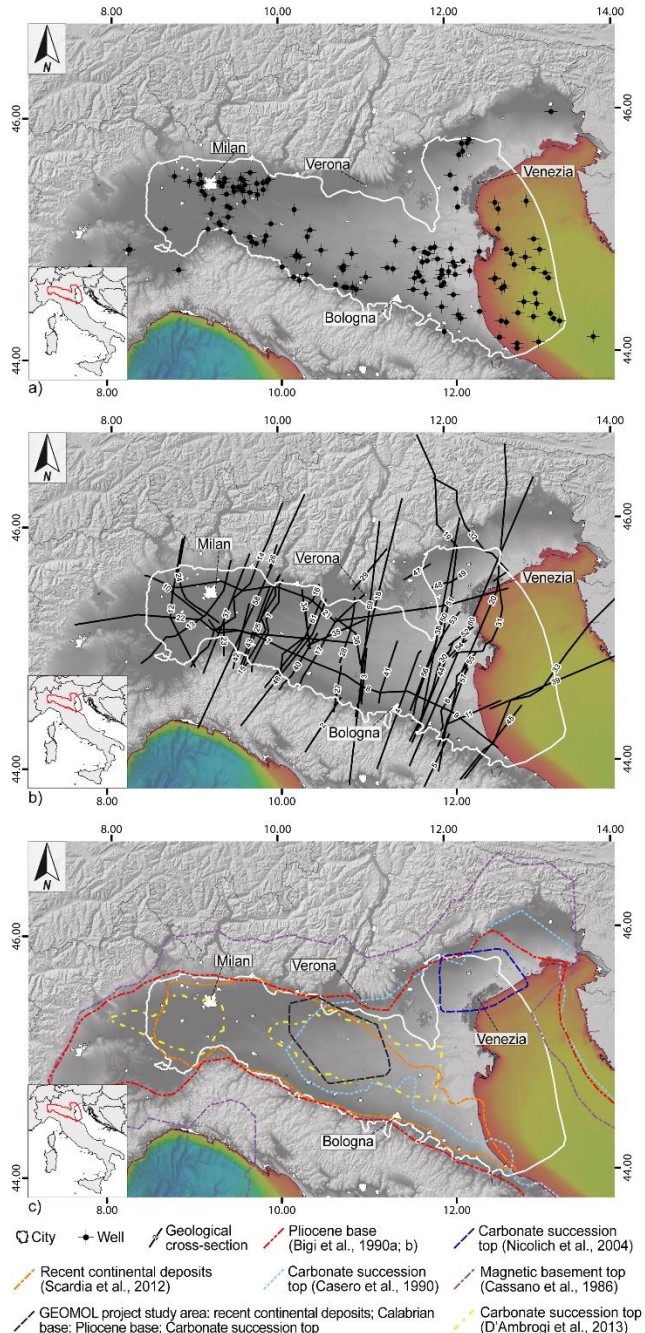

**Figure 4: Primitive data collected from public sources (i.e., databases and articles from the literature). a) Location of the 160 well data collected and digitized. Some areas of the Po Plain are characterized by a low well density, hence the successive primitive data collection (i.e., geological cross-sections and maps) were focused within a slightly reduced area indicated by the white polygon. b) Traces of the geological cross-sections collected from the literature (the number on each section corresponds to a specific geological cross-section in Table 1 where the data source is specified). c) Primitive data from geological subsurface maps. The source is reported in the legend.**

**Table 1 –** List and type of data collected for each lithological boundary.

| Lithological boundary | Data source | Data type |
|---|---|---|
| | ViDEPI Project | Well tops |
| | Boccaletti et al., 2011 | Geological cross-sections |
| | ISPRA, 2015 | Geological cross-sections |
| Base of recent continental deposits | ISPRA, 2015 | Isobath map |
| | Picotti et al., 2006 | Geological cross-sections |
| | Scardia et al., 2012 | Isobath map |
| | Wilson et al., 2009 | Geological cross-sections |
| | Wilson et al., 2009 | Geological cross-sections |
| | ViDEPI Project | Well tops |
| | Boccaletti et al., 2011 | Geological cross-sections |
| | Casero, 2004; | Geological cross-sections |
| | Cassano et al., 1986 | Geological cross-sections |
| | Fantoni and Franciosi, 2009 | Geological cross-sections |
| | ISPRA, 2015 | Geological cross-sections |
| | ISPRA, 2015 | Isobath map |
| Calabrian base | Lindquist, 1999 | Geological cross-sections |
| | Livani et al., 2018 | Geological cross-sections |
| | Maesano et al., 2015 | Geological cross-sections |
| | Picotti et al., 2006 | Geological cross-sections |
| | Pola et al., 2014 | Geological cross-sections |
| | Toscani et al., 2009 | Geological cross-sections |
| | Turrini et al., 2015 | Geological cross-sections |
| | Wilson et al., 2009 | Geological cross-sections |
| | ViDEPI Project | Well tops |
| | Bigi et al., 1990 | Isobath map |
| | Boccaletti et al., 2011 | Geological cross-sections |
| | Casero, 2004; | Geological cross-sections |
| | Cassano et al., 1986 | Geological cross-sections |
| Pliocene base | Fantoni & Franciosi, 2009 | Geological cross-sections |
| | ISPRA, 2015 | Geological cross-sections |
| | ISPRA, 2015 | Isobath map |
| | Livani et al., 2018 | Geological cross-sections |
| | Maesano et al., 2015 | Geological cross-sections |
| | Picotti et al., 2006 | Geological cross-sections |

| | | |
|---|---|---|
| | Pola et al., 2014 | Geological cross-sections |
| | Toscani et al., 2009 | Geological cross-sections |
| | Turrini et al., 2015 | Geological cross-sections |
| | Wilson et al., 2009 | Geological cross-sections |
| | ViDEPI Project | Well tops |
| | Boccaletti et al., 2011 | Geological cross-sections |
| | Casero, 2004; | Geological cross-sections |
| | Casero et al., 1990 | Isobath map |
| | Cassano et al., 1986 | Geological cross-sections |
| | Fantoni & Franciosi, 2009 | Geological cross-sections |
| | D'Ambrogi et al., 2023 | Isobath map |
| | ISPRA, 2015 | Geological cross-sections |
| | ISPRA, 2015 | Isobath map |
| Carbonate top | Lindquist, 1999 | Geological cross-sections |
| | Livani et al., 2018 | Geological cross-sections |
| | Maesano et al., 2015 | Geological cross-sections |
| | Nicolich, 2004 | Isobath map |
| | Picotti et al., 2006 | Geological cross-sections |
| | Pola et al., 2014 | Geological cross-sections |
| | Toscani et al., 2009 | Geological cross-sections |
| | Turrini et al., 2015 | Geological cross-sections |
| | Wilson et al., 2009 | Geological cross-sections |
| Magnetic basement top | Cassano et al., 1986 | Isobath map |

**Table 2 –** List of the geological cross-sections collected and used to build the 3D geological model. "Section number" refers to section traces reported in Fig. 4; "Figure number" refers to the figure in the original work; "Section" refers to the number of the section in the corresponding figure of the original work, "Repositioned" indicates whether the section trace has been modified according to the intersections with data located more accurately; "Section length" represents the length in kilometres of each geological sections. The total length of the collected geological cross-sections is 6341 km.


| Section number (present work) | Data source | Figure number (source work) | Section (source work) | Repositioned | Section length (km) |
|---|---|---|---|---|---|
| 1 | | Fig. 5 | A-A' | | 92.54 |
| 2 | | Fig. 5 | B-B' | ✓ | 59.3 |
| 3 | Boccaletti et al., 2011 | Fig. 5 | C-C' | | 50.6 |
| 4 | | Fig. 5 | D-D' | ✓ | 115.35 |
| 5 | | Fig. 5 | E-E' | ✓ | 53.16 |

| | | | | | |
|---|---|---|---|---|---|
| 6 | | Fig. 5 | F-F' | ✓ | 63.46 |
| 7 | | Fig. 5 | G-G' | | 108.46 |
| 8 | | Fig. 5 | H-H' | | 102.31 |
| 9 | | Fig. 5 | H'-H'' | | 88.82 |
| 10 | Casero, 2004 | Plate 2 | 2a | | 46.11 |
| 11 | | Plate 3 | 3b | ✓ | 97.24 |
| 12 | | | 3 | ✓ | 123.3 |
| 13 | | | 4 | ✓ | 131.36 |
| 14 | | | 5 | ✓ | 180.45 |
| 15 | | | 6 | ✓ | 103.58 |
| 16 | | | 7 | ✓ | 99.01 |
| 17 | Cassano et al., 1986 | | 8 | ✓ | 106.33 |
| 18 | | | 9 | ✓ | 205.48 |
| 19 | | | 10 | ✓ | 279.5 |
| 20 | | | 11 | ✓ | 289.43 |
| 21 | | | 12 | ✓ | 351.38 |
| 22 | | | 13 | ✓ | 267.05 |
| 23 | | Fig. 3 | 1(1) | ✓ | 30.94 |
| 24 | | Fig. 3 | 1(2) | ✓ | 96.03 |
| 25 | | Fig. 3 | 2(1) | ✓ | 86.48 |
| 26 | | Fig. 3 | 2(2) | ✓ | 21.54 |
| 27 | | Fig. 3 | 3(1) | ✓ | 42.23 |
| 28 | Fantoni and Franciosi, 2009 | Fig. 3 | 3(2) | ✓ | 61.17 |
| 29 | | Fig. 3 | 3(3) | ✓ | 48.03 |
| 30 | | Fig. 3 | 4 | ✓ | 92.45 |
| 31 | | Fig. 3 | 4 | ✓ | 98.25 |
| 32 | | Fig. 3 | 4 | ✓ | 131.64 |
| 33 | | Fig. 3 | 5 | ✓ | 238.39 |
| 34 | | | A-A' | | 54.11 |
| 35 | ISPRA, 2015 | | B-B' | | 57.99 |
| 36 | | | C-C' | | 52.23 |
| 37 | Lindquist, 1999 | Fig. 3a | | ✓ | 117.06 |

| | | | | | |
|---|---|---|---|---|---|
| 38 | | Fig. 3b | | ✓ | 194.67 |
| 39 | | Fig. 3c | | | 131.08 |
| 40 | Livani et al., 2018 | Fig. 11 | C-C' | | 62.4 |
| 41 | | Fig. 11 | D-D' | | 58.78 |
| 42 | | Fig. 6 | 1 | | 60.47 |
| 43 | Maesano et al., 2015 | Fig. 6 | 2 | | 78.36 |
| 44 | | Fig. 6 | 3 | | 99.78 |
| 45 | | Fig. 6 | 4 | | 84.22 |
| 46 | Picotti et al., 2007 | Fig. 5 | A-A' | | 140.53 |
| 47 | | Fig. 5 | A-A' | | 27,72 |
| 48 | | Fig. 5 | AA-AA'(1) | | 14.55 |
| 49 | | Fig. 5 | AA-AA'(2) | | 20.09 |
| 50 | | Fig. 5 | B-B'(1) | | 18.13 |
| 51 | Pola et al., 2014 | Fig. 5 | B-B'(2) | | 8.21 |
| 52 | | Fig. 5 | C-C' | | 27.35 |
| 53 | | Fig. 5 | CC-CC' | | 29.90 |
| 54 | | Fig. 5 | D-D' | | 21.10 |
| 55 | | Fig. 5 | E-E' | | 22.26 |
| 56 | Toscani et al., 2009 | Fig. 3 | A-A' | ✓ | 67.89 |
| 57 | | Fig. 3 | B-B' | ✓ | 67.74 |
| 58 | | Fig. 7 | B | | 235.47 |
| 59 | Turrini et al., 2015 | Fig. 7 | C | ✓ | 244.71 |
| 60 | | Fig. 7 | D | ✓ | 288.34 |
| 61 | Wilson et al., 2009 | Fig. 2c | | ✓ | 94.82 |
| | | | | **Total length (km)** | 6341.33 |

**Table 3 –** Wells collected in the database with their location (X-coordinate and Y-coordinate) and the rotary table elevation (m a.s.l.). The coordinates of wellhead locations are reported in the geographical system used in the database (WGS 84/UTM zone 32N; EPSG: 32632). UOI: well identification number. The deepest lithology unit reached by each well corresponds to:

(1) Recent continental deposits; (2) Late Pliocene-Pleistocene deposits; (3) Late Miocene-Late Pliocene deposits; (4) Early-Late Miocene deposits; (5) Triassic-Eocene carbonate units; (6) Crystalline basement.

| UOI | Well name | X (Well head) | Y (Well head) | Rotary table (m a.s.l.) | Deepest lithology unit |
|---|---|---|---|---|---|
| W001 | Adele 1 | 775896 | 4948539 | 26.0 | 3 |
| W002 | Adriana 1 | 813510 | 4935428 | 26.0 | 4 |
| W003 | Afrodite 1 | 776447 | 4949455 | 17.5 | 4 |

| W004 | Agnadello 1 | 540028 | 5032805 | 101.0 | 4 |
|------|-------------|--------|---------|-------|---|
| W005 | Albertina 1 | 817436 | 4975718 | 12.5 | 4 |
| W006 | Alex 1 | 814736 | 4925902 | 22.0 | 4 |
| W007 | Alma 1 | 811808 | 4913895 | 12.2 | 3 |
| W008 | Amelia 2Bis | 797531 | 4917463 | 21.2 | 3 |
| W009 | Anguilla 1 | 782121 | 4910090 | 18.3 | 3 |
| W010 | Antegnate 1 | 562955 | 5037036 | 114.0 | 4 |
| W011 | Antinea 1Bis | 796791 | 4889359 | 18.9 | 5 |
| W012 | Arcade 1 | 750637 | 5073687 | 65.0 | 5 |
| W013 | Arcobaleno 1 | 776213 | 5018899 | 23.4 | 4 |
| W014 | Arese 1 | 504947 | 5044530 | 170.2 | 4 |
| W015 | Arlecchino 1 | 779266 | 5013094 | 24.7 | 4 |
| W016 | Arluno 1 | 495098 | 5038326 | 155.5 | 4 |
| W017 | Azzura 1 | 784981 | 4941745 | 27.0 | 3 |
| W018 | Baggiovara 1 | 645422 | 4940921 | 65.2 | 4 |
| W019 | Ballan 1 | 735056 | 5044275 | 21.4 | 5 |
| W020 | Bando 7 | 725514 | 4949472 | 5.0 | 4 |
| W021 | Baricella 1 | 702411 | 4949324 | 12.2 | 3 |
| W022 | Baura 001 | 714740 | 4971856 | 9.4 | 4 |
| W023 | Bedeschi 1 dir | 727384 | 4922999 | 21.0 | 4 |
| W024 | Bedeschi 1 dirA | 727384 | 4922999 | 21.0 | 4 |
| W025 | Belgoioso 1 | 523877 | 4998221 | 79.0 | 4 |
| W026 | Bellaria Mare 1 | 780580 | 4894982 | 17.6 | 4 |
| W027 | Berillo 1 | 786889 | 4908968 | 18.9 | 3 |
| W028 | Bertolani 1 Dir | 640133 | 4940818 | 79.0 | 4 |
| W029 | Bevilacqua 1 | 677149 | 4958950 | 19.3 | 3 |
| W030 | Bosco Rosso 1 | 616314 | 4975044 | 32.5 | 3 |
| W031 | Brignano 2 | 551050 | 5046463 | 157.0 | 5 |
| W032 | Canopo 1 | 796699 | 4884893 | 19.0 | 5 |
| W033 | Cantoni 1 | 603577 | 4987473 | 36.0 | 4 |
| W034 | Cargnacco 1 | 827970 | 5102613 | 87.0 | 6 |
| W035 | Carmela 1 | 866844 | 4895571 | 27.0 | 2 |
| W036 | Cascina Buzzoni 1 | 723577 | 4967057 | 9.0 | 4 |

| W037 | Cascina Nuova 1 dir | 701791 | 4976442 | 16.0 | 5 |
|------|---------------------|--------|---------|------|---|
| W038 | Cascina San Francesco 1 | 737689 | 4963897 | 4.5 | 4 |
| W039 | Cascina San Pietro 1 dir | 542559 | 5032244 | 100.0 | 4 |
| W040 | Case Pinelli 1 | 657445 | 4950268 | 34.7 | 4 |
| W041 | Casello 1 dir | 593907 | 4970938 | 48.0 | 4 |
| W042 | Castano 1 | 481511 | 5043078 | 182.4 | 5 |
| W043 | Castel Gabbiano 1 | 557324 | 5036392 | 108.0 | 4 |
| W044 | Cerere 1 | 802948 | 4927606 | 26.0 | 3 |
| W045 | Cernusco 1 | 527759 | 5040908 | 136.0 | 4 |
| W046 | Cernusco 3 | 527060 | 5037627 | 123.0 | 4 |
| W047 | Chiosone 1 | 570184 | 4999305 | 45.8 | 4 |
| W048 | Cinzia 1 | 835893 | 4909975 | 26.0 | 4 |
| W049 | Claudia 1 | 822486 | 4954196 | 27.5 | 4 |
| W050 | Codevigo 1 | 741986 | 5014848 | 5.0 | 5 |
| W051 | Cona 2 | 709424 | 4964097 | 8.2 | 4 |
| W052 | Cornegliano 19 | 532686 | 5013752 | 89.0 | 4 |
| W053 | Correggio 33 | 636767 | 4957061 | 41.8 | 3 |
| W054 | Correggio 34 dir | 637237 | 4960570 | 39.0 | 3 |
| W055 | Correggio 35 dir | 637233 | 4960569 | 39.0 | 3 |
| W056 | Correggio 36 dir | 637231 | 4960569 | 39.0 | 3 |
| W057 | Correggio 37 dir | 637229 | 4960566 | 39.0 | 3 |
| W058 | Correggio 38 dir | 637224 | 4960564 | 39.0 | 3 |
| W059 | Correggio 39 dir | 637222 | 4960562 | 39.0 | 3 |
| W060 | Correggio 39 dirA | 637222 | 4960562 | 39.0 | 3 |
| W061 | Correggio 40 dir | 635457 | 4958855 | 41.0 | 3 |
| W062 | Corsico 1 | 506882 | 5029271 | 115.4 | 4 |
| W063 | Corte Mezzo 1 | 739452 | 4960575 | 6.0 | 3 |
| W064 | Corte Vittoria 1 | 735863 | 4976384 | 10.0 | 5 |
| W065 | Cusignana 1 | 745549 | 5073487 | 80.0 | 5 |
| W066 | Daniela 1 | 791415 | 4972060 | 25.0 | 4 |
| W067 | Dolo 1 dir | 740530 | 5031608 | 8.0 | 3 |
| W068 | Fabbrico 1 | 644606 | 4971873 | 23.5 | 3 |
| W069 | Ferrara 1 | 714006 | 4965211 | 13.0 | 5 |

| W070 | Filetto 1 | 745087 | 4912388 | 18.0 | 4 |
|------|-----------|--------|---------|------|---|
| W071 | Filetto 1 dirA | 745087 | 4912388 | 18.0 | 4 |
| W072 | Gallignano 2 | 563219 | 5030619 | 93.0 | 4 |
| W073 | Gandini 2 dir | 544340 | 5029732 | 96.7 | 3 |
| W074 | Gemma 1 | 830405 | 4912198 | 27.5 | 4 |
| W075 | Ghiara 2 dir | 593461 | 4969928 | 51.0 | 4 |
| W076 | Ginevra 1 | 782492 | 4939951 | 26.0 | 3 |
| W077 | Gisolo 1 | 580476 | 4959830 | 285.0 | 4 |
| W078 | Gladiolo 1 | 811922 | 4958195 | 27.0 | 4 |
| W079 | Glenda 1 | 825659 | 4949631 | 27.0 | 4 |
| W080 | Goro 1 | 761932 | 4974041 | 7.0 | 4 |
| W081 | Gudo Gambaredo 1 dir | 509669 | 5025876 | 111.0 | 2 |
| W082 | Inverno 1dir | 530799 | 5005318 | 84.1 | 4 |
| W083 | Irma 1 | 794204 | 4961737 | 27.0 | 4 |
| W084 | Isabella 1 | 811951 | 4985391 | 27.4 | 4 |
| W085 | Lanzano 1 | 529337 | 5026512 | 96.9 | 4 |
| W086 | Linarolo 1 | 522494 | 5000129 | 86.0 | 4 |
| W087 | Locate Triulzi 1 | 517074 | 5022007 | 103.6 | 3 |
| W088 | Maiero 1 | 727422 | 4956521 | 6.5 | 4 |
| W089 | Malossa 4 | 542687 | 5041078 | 130.4 | 5 |
| W090 | Malossa B Iniezione | 544749 | 5039426 | 130.0 | 4 |
| W091 | Mariangela 1 | 779126 | 4994676 | 26.0 | 3 |
| W092 | Marrara 1 | 706920 | 4955104 | 18.7 | 4 |
| W093 | Marzeno 41 | 729689 | 4900222 | 177.0 | 5 |
| W094 | Merlengo 1 | 746589 | 5066132 | 48.2 | 4 |
| W095 | Mirazzano 1 dir | 525371 | 5033901 | 114.0 | 3 |
| W096 | Molinella 1 | 710103 | 4946510 | 15.0 | 4 |
| W097 | Montalbano 21 | 704740 | 4954756 | 11.2 | 4 |
| W098 | Monte Acuto 1 dir | 532837 | 4991334 | 79.0 | 4 |
| W099 | Montecchi 1 | 629875 | 4943513 | 101.8 | 4 |
| W100 | Montecchio 1 | 720948 | 4977826 | 8.0 | 4 |
| W101 | Moretta 1 | 385425 | 4956518 | 269.2 | 6 |
| W102 | Muradolo 1 | 565668 | 4988208 | 48.0 | 4 |

| | | | | | |
|---|---|---|---|---|---|
| W103 | Negrini 1 | 726897 | 4942291 | 4.2 | 4 |
| W104 | Nervesa 1 | 752475 | 5076899 | 73.5 | 5 |
| W105 | Nervesa 1 dirA | 752475 | 5076899 | 73.5 | 4 |
| W106 | Novi Ligure 2 | 485442 | 4956778 | 192.5 | 3 |
| W107 | Offanengo 1 | 557914 | 5026164 | 86.0 | 3 |
| W108 | Oriana 1 | 827485 | 4916892 | 26.0 | 4 |
| W109 | Ornella 1 | 787040 | 4975930 | 18.9 | 4 |
| W110 | Paese 1 dir | 743131 | 5062880 | 39.0 | 4 |
| W111 | Pandino 1 | 536388 | 5029141 | 87.0 | 4 |
| W112 | Pavonara 1 | 711695 | 4973319 | 8.0 | 4 |
| W113 | Portoverrara 3 | 727680 | 4952637 | 4.6 | 4 |
| W114 | Priorato 1 | 592876 | 4969565 | 50.0 | 4 |
| W115 | Priorato 2 dir | 592876 | 4969565 | 50.0 | 3 |
| W116 | Pumenengo 1 | 566508 | 5038091 | 123.0 | 4 |
| W117 | Quarto 1 | 552990 | 4982890 | 89.0 | 4 |
| W118 | Rachele 1 | 805089 | 5020241 | 27.0 | 4 |
| W119 | Raffaella 2 | 804262 | 4968628 | 27.6 | 4 |
| W120 | Rea 1 dir | 512307 | 4994494 | 73.0 | 5 |
| W121 | Riccardina 1 | 701174 | 4938999 | 25.9 | 4 |
| W122 | Rolassa 1 | 440470 | 4975679 | 156.0 | 4 |
| W123 | Russi 1 dir | 742428 | 4915614 | 15.0 | 4 |
| W124 | Salerano 001 | 530085 | 5015436 | 83.4 | 4 |
| W125 | San Alessandro 1 | 596519 | 4968724 | 55.0 | 4 |
| W126 | San Alessandro 1 dirA | 596519 | 4968724 | 55.0 | 3 |
| W127 | San Cipriano 1 | 545609 | 5019541 | 72.9 | 3 |
| W128 | San Ermelinda 1 | 729376 | 4945094 | 2.0 | 3 |
| W129 | San Genesio 1 | 515135 | 5008502 | 95.5 | 5 |
| W130 | San Michele 1 | 596263 | 4947217 | 241.0 | 4 |
| W131 | San Polo 1 dir | 556581 | 4981094 | 93.0 | 4 |
| W132 | Sartirana 1 | 473042 | 4994514 | 108.0 | 4 |
| W133 | Scandiano 1 dirB | 637657 | 4941105 | 80.2 | 4 |
| W134 | Scandiano 2 dir | 639189 | 4940924 | 75.2 | 4 |
| W135 | Schiorsi 1 | 733784 | 4955454 | 4.3 | 3 |

| W136 | Segrate 1 | 522534 | 5037547 | 122.0 | 4 |
|------|-----------|--------|---------|-------|---|
| W137 | Seniga 1 | 591646 | 5012369 | 56.0 | 3 |
| W138 | Serena 1 | 816551 | 4890738 | 28.6 | 3 |
| W139 | Seresole 1 | 546169 | 5035239 | 112.0 | 4 |
| W140 | Sermide 1 | 684777 | 4983402 | 16.9 | 3 |
| W141 | Settimo Milanese 1 | 503716 | 5035857 | 146.5 | 5 |
| W142 | Solarolo 1 | 606267 | 4993664 | 43.0 | 5 |
| W143 | Sommariva Del Bosco 1 | 403848 | 4959735 | 307.7 | 4 |
| W144 | Spada 1 | 682501 | 4959755 | 24.9 | 4 |
| W145 | Torrazza 1 | 762257 | 4918726 | 6.0 | 2 |
| W146 | Torre Del Poggio 1 | 550408 | 4986992 | 75.0 | 4 |
| W147 | Torrente Riglio 1 dir | 564192 | 4983772 | 66.0 | 4 |
| W148 | Trava 1 | 739883 | 4952560 | 3.2 | 3 |
| W149 | Trenno 1 | 506511 | 5037331 | 146.0 | 4 |
| W150 | Trescore 1 | 550917 | 5028054 | 86.4 | 4 |
| W151 | Urago D'Oglio 1 | 568054 | 5040092 | 121.0 | 4 |
| W152 | Vaiano 1 | 535573 | 5033071 | 104.3 | 4 |
| W153 | Valgera 1 | 439138 | 4975331 | 131.0 | 4 |
| W154 | Valle Isola 1 | 753080 | 4956134 | 8.0 | 3 |
| W155 | Valletta 1 dir | 712816 | 4935159 | 15.0 | 3 |
| W156 | Varano 1 | 746924 | 4964965 | 8.5 | 4 |
| W157 | Vigatto 10 dir | 602968 | 4952521 | 132.2 | 4 |
| W158 | Vignola 1 | 717279 | 4977064 | 11.5 | 5 |
| W159 | Villavecchia 1 dir | 591224 | 4950301 | 237.2 | 4 |
| W160 | Zoboli 1 | 648649 | 4938952 | 70.0 | 4 |

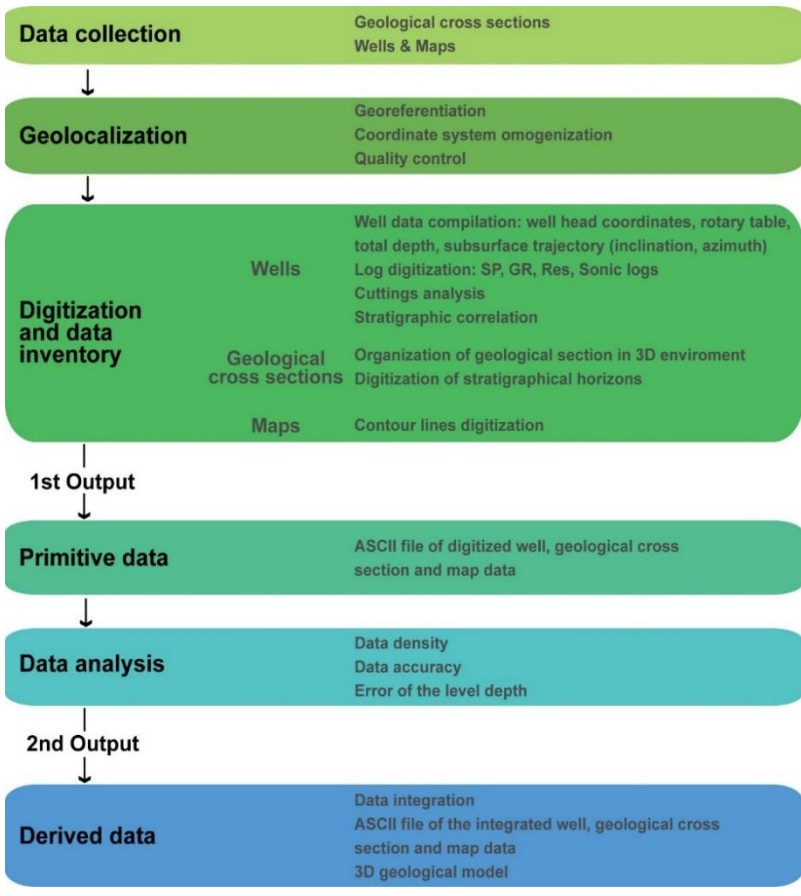

| | | |
|---|---|---|
| **Data collection** | | Geological cross sections<br>Wells & Maps |
| **Geolocalization** | | Georeferentiation<br>Coordinate system omogenization<br>Quality control |
| **Digitization and data inventory** | Wells | Well data compilation: well head coordinates, rotary table, total depth, subsurface trajectory (inclination, azimuth)<br>Log digitization: SP, GR, Res, Sonic logs<br>Cuttings analysis<br>Stratigraphic correlation |
| | Geological cross sections | Organization of geological section in 3D enviroment<br>Digitization of stratigraphical horizons |
| | Maps | Contour lines digitization |

**1st Output**

| | |
|---|---|
| **Primitive data** | ASCII file of digitized well, geological cross section and map data |
| **Data analysis** | Data density<br>Data accuracy<br>Error of the level depth |

**2nd Output**

| | |
|---|---|
| **Derived data** | Data integration<br>ASCII file of the integrated well, geological cross section and map data<br>3D geological model |


**Figure 5: Workflow used for the database creation.**

## 4 Primitive data: methods and results

Primitive data derive from the digitization of public data that have been graphically and spatially checked to eliminate errors due to low graphical quality and distortions, scale errors or bad positioning. We digitized the principal horizons reported in the
geological cross-sections, the isobaths of the main geological surfaces represented in the geological maps, and the well locations comprised their trajectory along depth, lithological and stratigraphical information, and geophysical logs from well composite logs. We performed the entire workflow (data collection, image georeferencing, data quality check, integration, and model-building) by using QGis and Petrel® software. The digitized data have been then organized in text files. Further details about the data processing are given below.

**4.1 Well data**

The source of well data is the VIDEPI project database (http://www.videpi.com). We collected data from 160 well logs, originally in a raster format (scale 1:1000), drilled in the Po Plain and in the northern Adriatic Sea (Fig. 4). Borehole

information, such as wellhead coordinates, rotary table elevation, measured depths, true depths, total depth, and deviation survey, is indicated on the composite logs, along with lithological, stratigraphic and fluid saturation information (Fig. 6). In
addition, composite logs include the Spontaneous Potential log (SP), which is used for lithological characterization and stratigraphic correlation purposes, and the Resistivity log (Res), used for the identification of hydrocarbon bearing intervals; in the more recent well master logs, the SP log is replaced by, or in certain cases complemented, a Gamma Ray log (GR). Furthermore, 133 out of 160 well composite logs of our dataset also include sonic log registrations that provide insights into sonic velocity variations with depth (Fig. 6). Further lithological information derives either from drill cuttings or from
laboratory analysis of core samples collected from wells.

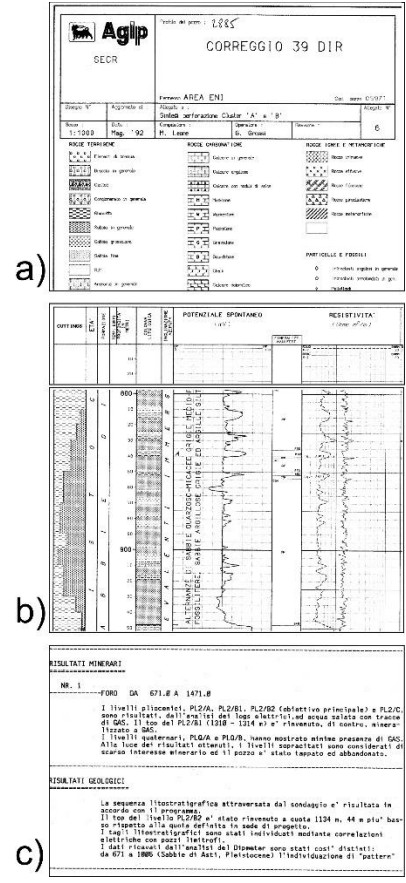

**Figure 6: Typical composite log (scale 1:1000) available from the VIDEPI database (www.videpi.com), originally in .pdf format. a) Section of the well name, well coordinates information and well log legend. b) Section of the well log data, lithological cuttings and completion information. c) Notes section (e.g., well trajectory, core information, geological information, technical data).**


An accurate revision of the wellhead positioning and the subsurface trajectory was necessary to properly collect and organize the well data and to furnish a dataset suitable to be imported into the most common 3D modelling software. From each well, we first transformed the reported geographic coordinates (expressed in ROMA 40 as geodetic datum) into projected

coordinates using the WEST BOAGA projection (geodetic datum: ROMA 40) and then in the geographical system used for
our database (i.e., WGS 84/UTM zone 32N; EPSG: 32632). Most of the wells are vertical and, hence, the well trajectory at
depth is set using the wellhead coordinates and the total depth values. On the contrary, the path of the directional wells was
reconstructed by using inclination (Inc) and azimuth (Az) information at the depth reported on the composite logs.

We then digitized the available SP log, GR log, and Sonic log of each well using the WebPlotDigitizer software (Rohatgi,
2014). Table 4 shows the log availability for each well in the project. The digitization procedure was performed manually,
with a variable sampling step, or by a semi-automatic method of line recognition. The digitized logs were then resampled to a
constant step of 0.5 m. Fig. 7 shows an example of the digitization process.

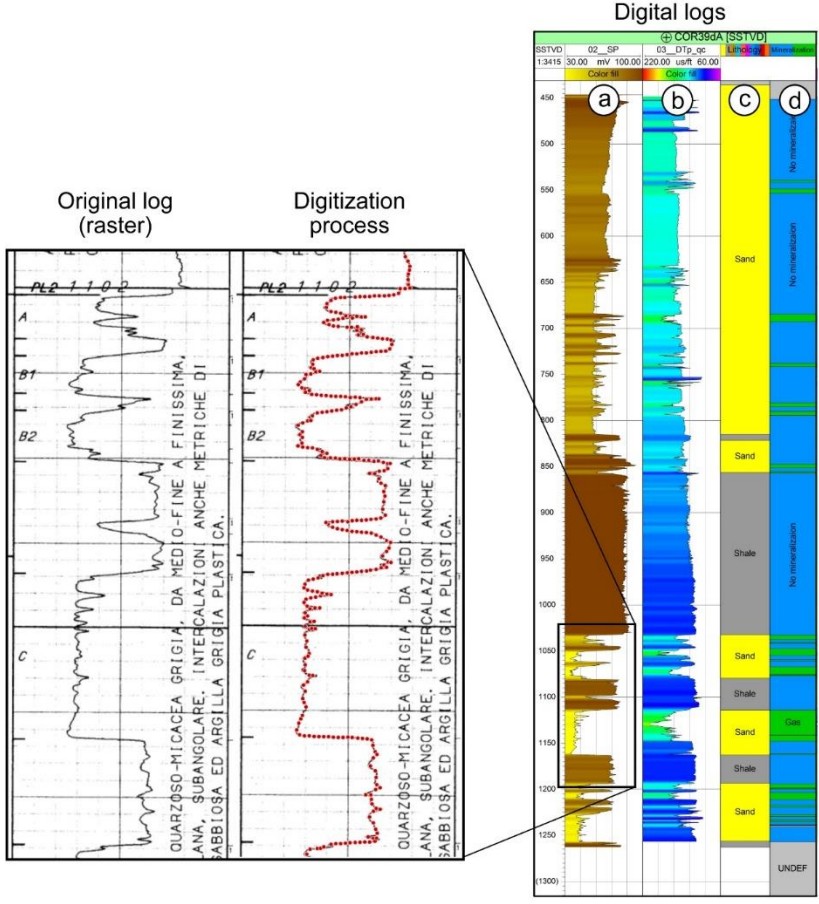

**Figure 7: Example of the digitization process for Well Correggio 39 dirA. The left panel shows the original Spontaneous Potential
log in the composite log (raster format) and the digitalized points (in red). The right panel shows the digitized (a) SP and (b) Sonic**
**logs and the classification of (c) Lithology and (d) Hydrocarbon bearing sections.**

**Table 4** – List of collected wells and the relative log availability. "UOI" indicates the identification number for each well. GR: Gamma Ray log; SP Spontaneous Potential log.

| UOI | Well name | GR | Lithology | Mineralization | Sonic | SP |
|---|---|---|---|---|---|---|
| W001 | Adele 1 | | ✓ | ✓ | ✓ | ✓ |
| W002 | Adriana 1 | | ✓ | ✓ | ✓ | ✓ |
| W003 | Afrodite 1 | | ✓ | ✓ | ✓ | ✓ |
| W004 | Agnadello 1 | | ✓ | ✓ | | ✓ |
| W005 | Albertina 1 | | ✓ | ✓ | ✓ | ✓ |
| W006 | Alex 1 | | ✓ | ✓ | | ✓ |
| W007 | Alma 1 | | ✓ | ✓ | | ✓ |
| W008 | Amelia 2Bis | | ✓ | ✓ | | ✓ |
| W009 | Anguilla 1 | | ✓ | ✓ | ✓ | ✓ |
| W010 | Antegnate 1 | | ✓ | ✓ | | ✓ |
| W011 | Antinea 1Bis | | ✓ | ✓ | | ✓ |
| W012 | Arcade 1 | | ✓ | ✓ | ✓ | ✓ |
| W013 | Arcobaleno 1 | ✓ | ✓ | ✓ | ✓ | |
| W014 | Arese 1 | | ✓ | ✓ | | ✓ |
| W015 | Arlecchino 1 | ✓ | ✓ | ✓ | ✓ | |
| W016 | Arluno 1 | | ✓ | ✓ | | ✓ |
| W017 | Azzura 1 | | ✓ | ✓ | ✓ | ✓ |
| W018 | Baggiovara 1 | | ✓ | ✓ | ✓ | ✓ |
| W019 | Ballan 1 | ✓ | ✓ | ✓ | ✓ | |
| W020 | Bando 7 | | ✓ | ✓ | ✓ | ✓ |
| W021 | Baricella 1 | | ✓ | ✓ | | ✓ |
| W022 | Baura 001 | | ✓ | ✓ | | ✓ |
| W023 | Bedeschi 1 dir | | ✓ | ✓ | ✓ | ✓ |
| W024 | Bedeschi 1 dirA | | ✓ | ✓ | ✓ | ✓ |
| W025 | Belgoioso 1 | | ✓ | ✓ | ✓ | ✓ |
| W026 | Bellaria Mare 1 | | ✓ | ✓ | | ✓ |
| W027 | Berillo 1 | | ✓ | ✓ | ✓ | ✓ |
| W028 | Bertolani 1 Dir | | ✓ | ✓ | ✓ | ✓ |
| W029 | Bevilacqua 1 | | ✓ | ✓ | ✓ | ✓ |
| W030 | Bosco Rosso 1 | | ✓ | ✓ | ✓ | ✓ |
| W031 | Brignano 2 | | ✓ | ✓ | ✓ | ✓ |
| W032 | Canopo 1 | | ✓ | ✓ | | ✓ |
| W033 | Cantoni 1 | | ✓ | ✓ | ✓ | ✓ |
| W034 | Cargnacco 1 | | ✓ | ✓ | ✓ | ✓ |
| W035 | Carmela 1 | | ✓ | ✓ | ✓ | ✓ |
| W036 | Cascina Buzzoni 1 | | ✓ | ✓ | ✓ | |

| Code | Name | | | | | |
|------|------|---|---|---|---|---|
| W037 | Cascina Nuova 1 dir | | ✓ | ✓ | ✓ | ✓ |
| W038 | Cascina San Francesco 1 | | ✓ | ✓ | ✓ | ✓ |
| W039 | Cascina San Pietro 1 dir | | ✓ | ✓ | ✓ | ✓ |
| W040 | Case Pinelli 1 | | ✓ | ✓ | ✓ | ✓ |
| W041 | Casello 1 dir | ✓ | ✓ | ✓ | ✓ | |
| W042 | Castano 1 | ✓ | ✓ | ✓ | ✓ | ✓ |
| W043 | Castel Gabbiano 1 | | ✓ | ✓ | ✓ | ✓ |
| W044 | Cerere 1 | | ✓ | ✓ | ✓ | ✓ |
| W045 | Cernusco 1 | | ✓ | ✓ | | ✓ |
| W046 | Cernusco 3 | | ✓ | ✓ | | ✓ |
| W047 | Chiosone 1 | | ✓ | ✓ | ✓ | ✓ |
| W048 | Cinzia 1 | | ✓ | ✓ | ✓ | ✓ |
| W049 | Claudia 1 | | ✓ | ✓ | ✓ | ✓ |
| W050 | Codevigo 1 | | ✓ | ✓ | ✓ | ✓ |
| W051 | Cona 2 | | ✓ | ✓ | | ✓ |
| W052 | Cornegliano 19 | | ✓ | ✓ | | ✓ |
| W053 | Correggio 33 | | ✓ | ✓ | ✓ | ✓ |
| W054 | Correggio 34 dir | | ✓ | ✓ | ✓ | ✓ |
| W055 | Correggio 35 dir | | ✓ | ✓ | ✓ | ✓ |
| W056 | Correggio 36 dir | | ✓ | ✓ | ✓ | ✓ |
| W057 | Correggio 37 dir | | ✓ | ✓ | ✓ | ✓ |
| W058 | Correggio 38 dir | | ✓ | ✓ | ✓ | ✓ |
| W059 | Correggio 39 dir | | ✓ | ✓ | ✓ | ✓ |
| W060 | Correggio 39 dirA | | ✓ | ✓ | ✓ | ✓ |
| W061 | Correggio 40 dir | | ✓ | ✓ | ✓ | ✓ |
| W062 | Corsico 1 | | ✓ | ✓ | | ✓ |
| W063 | Corte Mezzo 1 | | ✓ | ✓ | ✓ | ✓ |
| W064 | Corte Vittoria 1 | ✓ | ✓ | ✓ | ✓ | ✓ |
| W065 | Cusignana 1 | | ✓ | ✓ | ✓ | ✓ |
| W066 | Daniela 1 | | ✓ | ✓ | ✓ | ✓ |
| W067 | Dolo 1 dir | | ✓ | ✓ | ✓ | ✓ |
| W068 | Fabbrico 1 | | ✓ | ✓ | ✓ | ✓ |
| W069 | Ferrara 1 | | ✓ | ✓ | | ✓ |
| W070 | Filetto 1 | | ✓ | ✓ | ✓ | ✓ |
| W071 | Filetto 1 dirA | | ✓ | ✓ | ✓ | ✓ |
| W072 | Gallignano 2 | | ✓ | ✓ | | ✓ |
| W073 | Gandini 2 dir | | ✓ | ✓ | ✓ | ✓ |
| W074 | Gemma 1 | | ✓ | ✓ | ✓ | ✓ |
| W075 | Ghiara 2 dir | | ✓ | ✓ | ✓ | ✓ |

| | | | | | | |
|---|---|:-:|:-:|:-:|:-:|:-:|
| W076 | Ginevra 1 | | ✓ | ✓ | ✓ | ✓ |
| W077 | Gisolo 1 | | ✓ | ✓ | ✓ | ✓ |
| W078 | Gladiolo 1 | | ✓ | ✓ | ✓ | ✓ |
| W079 | Glenda 1 | | ✓ | ✓ | ✓ | ✓ |
| W080 | Goro 1 | | ✓ | ✓ | ✓ | ✓ |
| W081 | Gudo Gambaredo 1 dir | | ✓ | ✓ | ✓ | ✓ |
| W082 | Inverno 1dir | | ✓ | ✓ | ✓ | |
| W083 | Irma 1 | | ✓ | ✓ | ✓ | ✓ |
| W084 | Isabella 1 | | ✓ | ✓ | ✓ | ✓ |
| W085 | Lanzano 1 | | ✓ | ✓ | | ✓ |
| W086 | Linarolo 1 | | ✓ | ✓ | ✓ | ✓ |
| W087 | Locate Triulzi 1 | | ✓ | ✓ | | ✓ |
| W088 | Maiero 1 | | ✓ | ✓ | | ✓ |
| W089 | Malossa 4 | ✓ | ✓ | ✓ | | ✓ |
| W090 | Malossa B Iniezione | | ✓ | ✓ | ✓ | ✓ |
| W091 | Mariangela 1 | | ✓ | ✓ | ✓ | ✓ |
| W092 | Marrara 1 | | ✓ | ✓ | | ✓ |
| W093 | Marzeno 41 | | ✓ | ✓ | ✓ | ✓ |
| W094 | Merlengo 1 | ✓ | ✓ | ✓ | ✓ | |
| W095 | Mirazzano 1 dir | | ✓ | ✓ | ✓ | ✓ |
| W096 | Molinella 1 | | ✓ | ✓ | ✓ | ✓ |
| W097 | Montalbano 21 | | ✓ | ✓ | | ✓ |
| W098 | Monte Acuto 1 dir | | ✓ | ✓ | ✓ | ✓ |
| W099 | Montecchi 1 | | ✓ | ✓ | ✓ | ✓ |
| W100 | Montecchio 1 | | ✓ | ✓ | ✓ | ✓ |
| W101 | Moretta 1 | | ✓ | ✓ | ✓ | ✓ |
| W102 | Muradolo 1 | | ✓ | ✓ | ✓ | ✓ |
| W103 | Negrini 1 | | ✓ | ✓ | ✓ | ✓ |
| W104 | Nervesa 1 | ✓ | ✓ | ✓ | ✓ | |
| W105 | Nervesa 1 dirA | ✓ | ✓ | ✓ | ✓ | |
| W106 | Novi Ligure 2 | | ✓ | ✓ | ✓ | ✓ |
| W107 | Offanengo 1 | | ✓ | ✓ | ✓ | ✓ |
| W108 | Oriana 1 | | ✓ | ✓ | ✓ | ✓ |
| W109 | Ornella 1 | | ✓ | ✓ | ✓ | ✓ |
| W110 | Paese 1 dir | ✓ | ✓ | ✓ | ✓ | |
| W111 | Pandino 1 | | ✓ | ✓ | | ✓ |
| W112 | Pavonara 1 | | ✓ | ✓ | ✓ | ✓ |
| W113 | Portoverrara 3 | | ✓ | ✓ | | ✓ |
| W114 | Priorato 1 | | ✓ | ✓ | ✓ | ✓ |

| | | | | | | |
|---|---|:---:|:---:|:---:|:---:|:---:|
| W115 | Priorato 2 dir | | ✓ | ✓ | ✓ | ✓ |
| W116 | Pumenengo 1 | | ✓ | ✓ | ✓ | ✓ |
| W117 | Quarto 1 | ✓ | ✓ | ✓ | ✓ | |
| W118 | Rachele 1 | | ✓ | ✓ | ✓ | ✓ |
| W119 | Raffaella 2 | | ✓ | ✓ | ✓ | ✓ |
| W120 | Rea 1 dir | | ✓ | ✓ | ✓ | ✓ |
| W121 | Riccardina 1 | | ✓ | ✓ | ✓ | ✓ |
| W122 | Rolassa 1 | | ✓ | ✓ | ✓ | ✓ |
| W123 | Russi 1 dir | | ✓ | ✓ | ✓ | ✓ |
| W124 | Salerano 001 | | ✓ | ✓ | | ✓ |
| W125 | San Alessandro 1 | | ✓ | ✓ | ✓ | ✓ |
| W126 | San Alessandro 1 dirA | | ✓ | ✓ | ✓ | ✓ |
| W127 | San Cipriano 1 | | ✓ | ✓ | | ✓ |
| W128 | San Ermelinda 1 | | ✓ | ✓ | ✓ | ✓ |
| W129 | San Genesio 1 | ✓ | ✓ | ✓ | ✓ | ✓ |
| W130 | San Michele 1 | | ✓ | ✓ | ✓ | ✓ |
| W131 | San Polo 1 dir | | ✓ | ✓ | ✓ | ✓ |
| W132 | Sartirana 1 | | ✓ | ✓ | ✓ | ✓ |
| W133 | Scandiano 1 dirB | | ✓ | ✓ | ✓ | ✓ |
| W134 | Scandiano 2 dir | | ✓ | ✓ | ✓ | ✓ |
| W135 | Schiorsi 1 | | ✓ | ✓ | ✓ | ✓ |
| W136 | Segrate 1 | | ✓ | ✓ | | ✓ |
| W137 | Seniga 1 | | ✓ | ✓ | ✓ | ✓ |
| W138 | Serena 1 | | ✓ | ✓ | ✓ | ✓ |
| W139 | Seresole 1 | | ✓ | ✓ | ✓ | ✓ |
| W140 | Sermide 1 | | ✓ | ✓ | ✓ | ✓ |
| W141 | Settimo Milanese 1 | | ✓ | ✓ | ✓ | ✓ |
| W142 | Solarolo 1 | | ✓ | ✓ | ✓ | ✓ |
| W143 | Sommariva Del Bosco 1 | ✓ | ✓ | ✓ | ✓ | ✓ |
| W144 | Spada 1 | | ✓ | ✓ | | ✓ |
| W145 | Torrazza 1 | ✓ | ✓ | ✓ | ✓ | ✓ |
| W146 | Torre Del Poggio 1 | | ✓ | ✓ | ✓ | ✓ |
| W147 | Torrente Riglio 1 dir | | ✓ | ✓ | ✓ | ✓ |
| W148 | Trava 1 | | ✓ | ✓ | ✓ | |
| W149 | Trenno 1 | ✓ | ✓ | ✓ | | ✓ |
| W150 | Trescore 1 | | ✓ | ✓ | | ✓ |
| W151 | Urago D'Oglio 1 | | ✓ | ✓ | ✓ | ✓ |
| W152 | Vaiano 1 | | ✓ | ✓ | ✓ | ✓ |
| W153 | Valgera 1 | | ✓ | ✓ | ✓ | ✓ |

| W154 | Valle Isola 1 | ✓ | ✓ | ✓ | ✓ |
| W155 | Valletta 1 dir | ✓ | ✓ | ✓ | ✓ |
| W156 | Varano 1 | ✓ | ✓ |  | ✓ |
| W157 | Vigatto 10 dir | ✓ | ✓ | ✓ | ✓ |
| W158 | Vignola 1 | ✓ | ✓ | ✓ | ✓ |
| W159 | Villavecchia 1 dir | ✓ | ✓ | ✓ | ✓ |
| W160 | Zoboli 1 | ✓ | ✓ | ✓ | ✓ |

The Resistivity log (Res) was used to identify the mineralized intervals of wells with gas bearing layers and then to create a new "discrete" type of log with indications regarding "hydrocarbon bearing" and "water bearing" layers. Subsequently, Res was used to isolate and remove the sonic log measurements for those geological intervals affected by the hydrocarbon presence (Figs. 7 and 8).

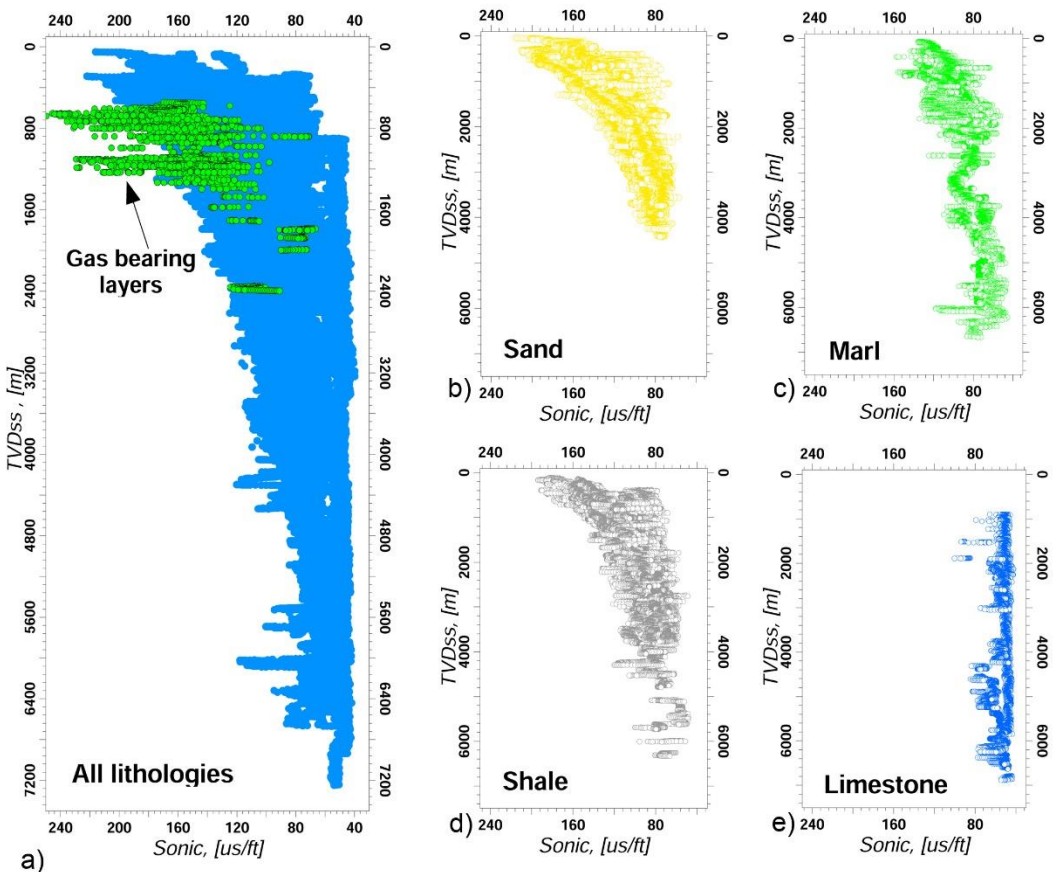

**Figure 8: a) Variation of transit time with depth for all samples showing the effect of the presence of gas. b-e) Variation of transit time with depth for the sand, marl, shale, and limestone lithologies, respectively.**

The stratigraphic information on the composite logs together with SP and GR logs was used to perform a correlation at the
regional scale by identifying those units showing different lithological properties and defining surfaces dividing geological
successions with different mechanical properties. . The main units recognized include (from youngest to oldest):

- Recent clastic deposits of the Po Plain and the Adriatic Sea, consisting of gravels and sands of continental
  environment identified from the cuttings of the first tens to hundreds of meters for each well. This unit has been
  widely analysed in the literature (Regione Emilia Romagna & ENI-AGIP, 1998; Regione Lombardia & ENI-
AGIP, 2002; Scardia et al., 2012; Ghielmi et al., 2013).
- Late Pliocene-Pleistocene sand-rich sequences, consisting of sands and clayey sands with clayey interlayers of deep
  marine to continental environment (Sabbie di Asti Group), connected to the latest filling of the Po Plain foredeep.
- Late Miocene-late Pliocene clastic deposits connected to the evolution of the northern Apennine foredeep and top-
  thrust basins, including:
i. clay-rich units made up of variably silty clays with minor sand (Argille del Santerno Fm.);
       ii. sand-rich (mostly foredeep turbiditic) units made up of thick sand beds with minor clay and thin-bedded
           sand-clay repetitions (e.g., Marnoso-Arenacea Fm., Bagnolo Fm., Fusignano Fm., Caviaga Fm., Porto
           Garibaldi Fm., Porto Corsini Fm.);
       iii. conglomeratic units made up of shallow-marine and fluvio-deltaic conglomerates and sands with minor clay
(e.g., Sergnano Fm., Cortemaggiore Fm., Boreca Conglomerate).
- Early-late Miocene marly sequences, consisting of marl, clayey marl, clay and sandy marl with sand intercalations
  recording deposition on the foreland ramp (e.g., Marne di Gallare Group).
- Triassic to Eocene undifferentiated carbonate units consisting of prevailing limestone (mainly mudstone and
  packstone-grainstone) and dolomite with subordinate marl.
- Variscan crystalline basement.

We collected additional data related to the lithological characteristics of the subsurface sedimentary units from cuttings
description. The cuttings description contains the information collected during mud logging, where rock fragments from the
borehole reach the surface due to the circulation of the drilling fluids. Those data were combined with the SP and GR logs and
with lithological data from core sample analysis reported in the well profile to characterize the lithology of the entire well. In
total, we identified 9 macro-lithologies listed below together with the descriptions commonly found in the well profiles:

- Gravel (e.g., polygenic gravel, gravel prevalent, polygenic gravel and sand with shale interbeds, gravel and pebbles
  with sand interbeds).
- Sand (e.g., sand, sand prevalent, shaly sand, fine sand, sand with shale interbeds, sand and shaly sands, sand banks).
- Cemented Sand (e.g., cemented sand, fine-grained cemented sand, cemented sand and pebbles, calcareous cemented
sand, sand of variable cementation).
- Shale (e.g., shale, silty shale, gray shale, marl shale, prevalent shale).
- Sand/shale alternances (e.g., gray shales and sand, shales with sand interbeds).

- Conglomerates (e.g., polygenic conglomerate, polygenic conglomerate with shale, polygenic conglomerate with sand interbeds).
- Marl (e.g., marls, silty marls, gray silty marls, marls and sandy marls, grey marls with cemented sand).
- Dolomite (e.g., dolomite, calcareous dolomite, crystalline dolomite, shaly dolomite, gray dolomite).
- Limestone (mudstone/wackestone/packstone/grainstone).

Well data represent the main constrain for the subsequent 3D geological modelling phase. As shown in Figure 4, the distribution of the wells in the area is not homogeneous and some regions are characterized by low well density. For this

reason, the collection of the other primitive data (i.e., geological cross-sections and maps) was focused in the area indicated by the white polygon in Figure 4, excluding the most isolated wells.

We performed a preliminary analysis of the sonic velocity variations with depth using the collected data and the newly created lithological and mineralization logs (Fig. 8). The sonic logs display transit time values in the range 40-140 μsec/ft. However, we observed even higher values (approx. 150-200 μsec/ft) in shallow formation that were concentrated to the first tens to few

hundred meters from the surface and are mostly connected to gravel lithologies that are characterized by poor consolidation (Fig. 8). In some cases, unusually high or low sonic log values with respect to the general data trend for a certain lithology can be ascribed to the presence of thin layers with different lithological characteristics, to possible borehole damage or to the presence of fractures. The currently available information does not allow a clear interpretation of their causes and even their removal. Most of the lithologies show a gradual decrease in transit time with increasing depth (Fig. 8b-d) and, at about 4 km

depth, the transit time flattens out showing rather constant values for higher depths, independently from the lithology. The continuous decrease of the P-waves transit time with depth reflects the increasing compaction of the sediments due to the overburden weight. The limestone is an exception, showing a relatively constant transit time independently of the depth. The presence of gas significantly increases the transit time with respect to water-saturated rocks: the so-called "gas effect" outlines as a rock density reduction (Fig. 8a).

The abundance of each lithology within the main units, defined in this study, significantly affects the average mechanical rock properties of the entire unit (Fig. S2). One of the advantages of the analysis performed is the possibility to assign specific mechanical properties for each lithology and for different depths. Knowing the amount of each lithology present in the geological units we can achieve a more precise characterization of the subsurface geological layers that is fundamental for improving the quality of the geomechanical simulations. Applications of the sonic log analysis in defining mechanical rock

properties of the subsurface of the Po Plain area can be found in Benetatos et al. (2023a, 2023b). In the first work the authors propose a workflow for geomechanical simulations through seabed monitoring. A significant step in the workflow involves the rock mechanical characterization that is performed through the analysis of the sonic log data that are converted to dynamic Young's modulus values specific for the different lithologies and for various depths. The same approach is also followed by Benetatos et al.(2023b) to calculate mechanical properties of the Argille del Santerno Fm.. They used well log data and

compared them to those derived from laboratory analysis of core samples. This comparative analysis revealed relations

between well log and laboratory derived properties and contributed to the understanding of the deformation behaviour of this important geological formation that extends across northern Italy.

## 4.2 Geological cross-sections

We collected 6341 km of published geological cross-sections (Cassano et al., 1986; Lindquist, 1999; Casero, 2004; Picotti et
al., 2006; Fantoni and Franciosi, 2009; Toscani et al., 2009; Wilson et al., 2009; Boccaletti et al., 2011; Pola et al., 2014; ISPRA, 2015; Maesano et al., 2015; Turrini et al., 2015; Livani et al., 2018; Table 1). Several procedures were implemented for uploading, calibrating, and revising the geological cross-sections in a 3D environment and, finally, for digitizing the selected horizons. The location maps (i.e., maps displaying the section traces) and the cross-sections were graphically re-arranged and improved to reduce imperfections and errors due to the low quality and/or images distortion. We then
georeferenced the location maps in QGis environment digitizing the relative section traces (Fig. 4). Based on the geographically oriented section traces, the cross-section images (raster format) were properly uploaded in a 3D environment. The geological cross-sections composed of segments with different orientations were cut into several parts and separately imported. Where necessary (i.e., location inconsistency), the geological cross-sections were repositioned using the intersections with other data such as wells, surface geology (e.g., geological boundaries, faults, etc.), other geological cross-sections, and orographic and
hydrographic features (e.g., rivers).

The revised and georeferenced cross-sections were finally uploaded into a 3D project in the Petrel® software. We digitized four geological horizons that roughly correspond to the boundaries of the main lithological discontinuities recognized through the well data analysis. The horizons are, from the oldest to the youngest: the top of the carbonate succession (which becomes more marly in its upper portion) dividing a mainly carbonate succession (below) by a mainly siliciclastic succession (above)
and it does not represent a chronostratigraphic boundary, the Pliocene base, the Calabrian base, and the base of recent continental deposits. In the regions characterized by a vertical duplication of the same unit due to the effect of thrusting, we digitized the hanging wall of the units until the hanging wall cut-off, then passing directly to the footwall of the same unit. In such a way, a marked artificial step was generated; however, we deem this approximation as necessary for the successive 3D modelling dataset creation, which does not integrate any fault element. In the data collection process, some public geological
cross-sections were excluded from the database and the digitization process. For instance, the AGIP geological cross-sections reported on the Italian geological maps at 1:100,000 scale (sheets 75, 77 and 88; https://www.isprambiente.gov.it/it/attivita/suolo-e-territorio/cartografia/carte-geologiche-e-geotematiche/carta-geologica-alla-scala-1-a-100000) were not used due to the geological interpretation based on scarce deep data information (many deep wells and seismic reflection profiles were unavailable at that time), strongly differing from interpretations recently proposed
on the basis of a large amount of subsoil data. Furthermore, some vintage sections (i.e., Bally et al., 1986) were excluded since some more recent geological cross-sections passing close to their traces show a more precise localization and better graphical properties allowing a more accurate digitization process.

For each digitized horizon a delimited text file in ASCII format reporting the xyz coordinates is generated. Since the digitization process was performed manually, the data are provided with an irregular sampling step.

### 4.3 Geological maps

The 3D database also includes data derived from 10 published subsurface geological maps. We digitized the following maps:

- 2 isobath maps of the base of the recent continental deposits (QC1 horizon map by ISPRA, 2015; Scardia et al., 2012);
- 1 isobath map of the Calabrian base (QM1 horizon map by ISPRA, 2015);
- 2 isobath maps of the Pliocene base (Bigi et al., 1990a, 1990b; ISPRA, 2015);
- 4 isobath maps of the carbonate succession top (Casero et al., 1990; Nicolich, 2004; ISPRA, 2015; D'Ambrogi et al., 2023);
- 1 isobath map of the magnetic basement top (Cassano et al., 1986).

The maps, available in hard copy and/or in raster format, were graphically revised and re-arranged in order to reduce possible scanning defects and distortions. The maps were then georeferenced, and the contour lines digitized by using QGis software. The digitized contour lines were exported and uploaded into the 3D Petrel® database. We then verified in a 3D environment their consistency with other subsurface geological information, especially from well data.

For each map, a delimited text file in ASCII format reporting the xyz coordinates of the contour lines was generated. Since the digitization process was performed manually, the data are provided with an irregular sampling step.

### 5 Data accuracy

A widespread accepted standard approach to address map accuracy is still not available. Recent studies suggest methods and standards for error analysis of geological or subsurface maps (e.g., Kint et al., 2020 and references therein). However, these methods are still affected by significant bias as they depend on the data availability and the criteria of data selection. Quality flagging is the basic approach used to quantify uncertainty within a spatial dataset and is done by assessing metadata fields. This method can be limited to indicate the presence or absence of data or be very complex producing a full range of quantitative error ranges (e.g., Bardossy and Fodor, 2001). Kint et al. (2020) presented an approach to assess data uncertainty for a well dataset in the Quaternary succession of the Belgian Continental Shelf. They produced confidence maps based on datasets from different origins and time periods. Their method consists in: i) determination of the data density (how much data contribute to each grid cell to provide information on lateral and depth variability); ii) direct mapping of measured errors and accuracies; iii) transformation of the measured values or categorical quality flags into uncertainty percentages; iv) selection of data subsets based on the uncertainty maps. Not all these points are always feasible or necessary. This renders the method non-general. For example, in the case of few data or datasets without associated uncertainty, steps ii) and iv) are not recommended/feasible.

Furthermore, the uncertainty drags errors due to instrumental (absolute accuracy, positioning accuracy) and human (expert judgment, data selection, data origin and representation) accuracy are not often quantifiable.

The only "universal" and "dataset unrelated" rule when considering the geographic space comes from the first law of geography (Tobler, 1970) that states: "everything is related to everything else, but near things are more related than distant things." This law is the foundation of the concepts of spatial dependence and spatial auto-correlation and is used specifically for the inverse distance weighting method for spatial interpolation (Shepard, 1968).

Generalizing the approach proposed by Kint et al. (2020) with application to arbitrary spatial data and using the Inverse
Distance Weight (IDW) for our analysis, we implemented, in a preliminary way, a method to weight the accuracy/confidence of geological surfaces. For each horizon we quantified: i) the data density contributing to assess the lateral accuracy and the depth variability, ii) the accuracy based on the data density and spatial autocorrelation converted into a probability (Inverse Distance Weight - IDW) describing the confidence on the data at each point of the study area and iii) the error associated with the depth of the geological surfaces due to discrepancies between the data of different origins where different guesses exist.
Hence, for each point of the study area, we provided a value indicating the accuracy and a value indicating the estimated error of the depth of the geological surface.

To apply our model, we started by edging the study area including the observation region and covering it through an evaluation grid (Fig. 9a). In surface and subsurface mapping, the observations come from properly identified sites. Locations of these sites (hereafter checkpoints) are known and tag the observations. The variable of interest (i.e., horizon depth) exists in every
point of the region (i.e., grid nodes) but is observed only in a finite set of locations (checkpoints). The variability model describes how uncertainties increase moving away from the checkpoint with respect to the best guess, built according to the observations and the geological constraints. In our case study, the properly identified sites are points (wells), lines (the traces of geological cross-sections digitized at discrete points), and polygons (here intended as raster fields digitized at discrete points, i.e., isobaths of geological maps). Note that not all kinds of checkpoints can be considered at the same level of confidence
since well data give better depth-constraints to those derived by geological cross-sections or interpolated maps. For this reason, it is convenient to assign higher specific weight to first-order checkpoints (well data in our case) with respect to higher order checkpoints (sections or maps). Once the hierarchical subdivision has been made, the inverse distance (ID) principle is used to model the uncertainties at each point of the study area (i.e., at grid nodes) based on spatial autocorrelation with respect to checkpoints as:

$$ID = \frac{1}{1 + r^p}$$

where $ID$ is the inverse distance, r is the distance between the point of observation and the checkpoint and p is the checkpoint order; in this study p=1 for well data, p=2 for sections, and p=3 for maps. The inverse distance ($ID$) is than normalized to obtain the inverse distance weight $IDW$ as:

$$IDW = \frac{ID - ID_{min}}{ID_{max} - ID_{min}}$$

The *IDW* assigns a confidence based on distance autocorrelation that decreases more gently when considering first order checkpoints (red curve goes under 0.2 at 7 km of distance; Fig. 9b) with respect to higher order checkpoints (green and black curves goes under 0.2 at 2 km of distance; Fig. 9b). The total weight calculated at each point of the study area is the mean value of the weights calculated with respect to different order checkpoints. This means that *IDW*=1 is the best guess assigned to grid nodes that, in the ideal case, lie - at the same time - over checkpoints of order 1 to n (i.e., in our case over a well crossed

by a section and coinciding with an interpolation point of the map). Statistics of the dataset are reported in Table 5.

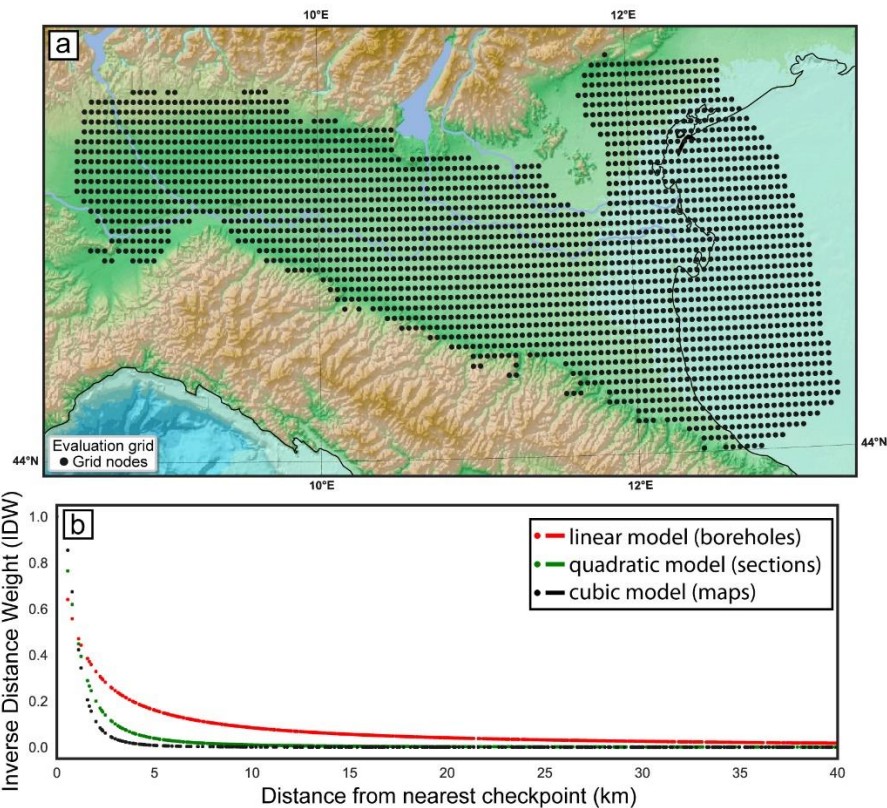

**Figure 9: a) Grid-nodes used to evaluate the data accuracy and uncertainties for geological surfaces presented for the study area. At each node, based on spatial autocorrelation, we calculated the Inverse Distance Weight (IDW) with respect to points observation (checkpoints). b) The model adopted for the IDW describes how uncertainties increase moving away from the best guess at**
**checkpoints (i.e., at well, along geologic cross-sections or on depth maps); at the same distance from a checkpoint, well data gives higher IDWs (red points) with respect to data derived from geological cross-sections or interpolated maps (green and black points).**

### 5.1 Data density

    Four out of the five processed geological surfaces (i.e., the base of recent continental deposits,  the Calabrian base, the Pliocene base and the Carbonate succession top) were reconstructed using checkpoints of order 1 (i.e., well data), 2 (i.e., geological
cross-sections) and 3 (i.e., subsurface geological maps). The reconstruction of the magnetic basement top derives from a unique source (Cassano et al., 1986) and, for this reason, we avoided to include this surface in the data accuracy analysis.

The analysis of our dataset (Table 5) shows that the three shallowest surfaces (i.e., recent continental deposits, Pleistocene, and Pliocene bases) are the most constrained with up to 139 checkpoints of the first order, whereas the Carbonate succession top is constrained by few checkpoints of order 1. The carbonate succession top and the Pliocene base are described by the highest amount of data with a density larger than 15 total checkpoints to each grid cell.

**Table 5** – Parameters of each unit. Checkpoint refers to the number of locations where the data are observed. P is the checkpoint order where p=1 for well data, p=2 for sections, and p=3 for maps.

| Lithological surface | Grid points | n. checkpoints (p1) | n. checkpoints (p2) | n. checkpoints (p3) | n. checkpoints (total - Pts. Density) |
|---|---|---|---|---|---|
| Recent continental deposits base | 2334 | 42 | 634 | 3082 | 3758 – 1.61 |
| Calabrian base | 2334 | 139 | 3065 | 2299 | 5503 – 2.36 |
| Pliocene base | 2334 | 114 | 4238 | 29795 | 34174 – 14.63 |
| Carbonate top | 2334 | 17 | 3309 | 169785 | 173111 – 74.17 |

**5.2 Data analysis**

The total number of checkpoints does not always increase the confidence level of the data. In fact, for the Calabrian base, the Pliocene base, and the Carbonate succession top, the *IDW* values also increase up to double when calculated considering checkpoints of all orders, whereas they decrease for recent continental deposits (dashed lines in Fig. 10). In the latter case, higher confidence is obtained when considering only the order 1 checkpoints (solid lines in Fig. 10). The best-constrained surface is confirmed to be the Pliocene base with a high number of both well data (i.e., order 1 checkpoints) and total data. The spatial distribution of the *IDW* for the proposed surfaces is reported in Fig. 11 and represents the confidence on the variable of interest (level depth) at each node of the grid (1 = max confidence). All the maps show maximum *IDW* ≈0.7 and a mean value of 0.06 (recent continental deposits – Fig. 11a), 0.09 (Calabrian base – Fig. 11b), 0.14 (Pliocene base – Fig. 11c) and 0.08 (Carbonate Top – Fig. 11d). The *IDW* values indicate that the Pliocene surface has a confidence level almost double than that obtained for the other surfaces on average. Interactive figures (html format to be opened in a web browser) of each map reported in Figure 11 are available in the supplementary material (S1.zip) and provide detailed statistical information at each node of the interpolation grid.

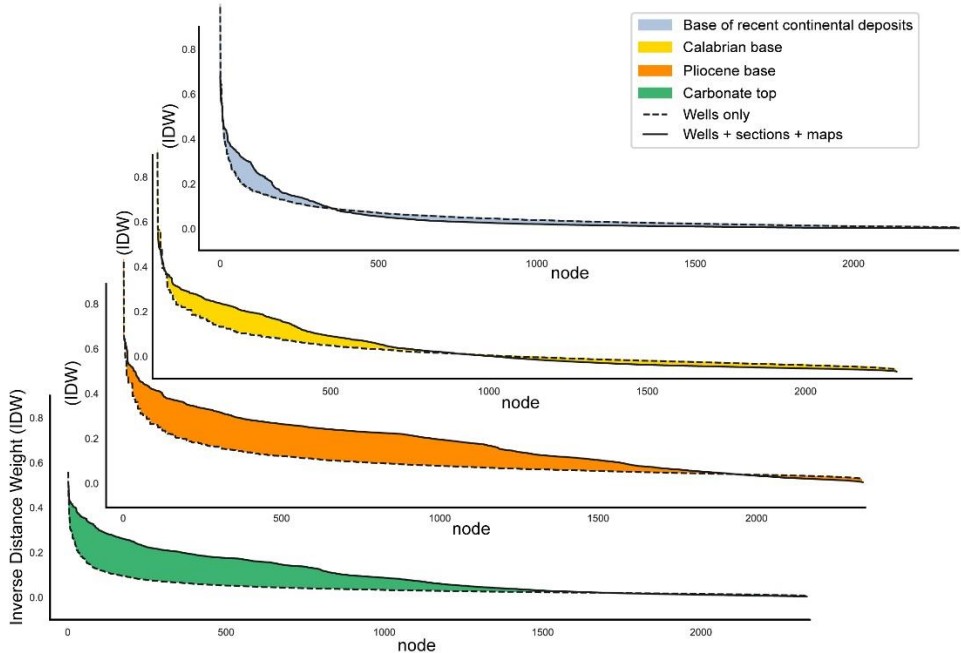

**Figure 10: IDW values at grid nodes calculated for each surface considering distances from well data only (dashed lines) or integrating distances from well data plus geological cross-section plus geological maps (solid lines). The area between the two lines for each proposed surface is larger for greater uncertainties. Note that the uncertainties increase with the depth of the surfaces and that the total number of checkpoints does not always increase the confidence level of the data (as for example in the recent continental deposits).**

### 5.3 Uncertainty associated with depth

The four surfaces analysed in this work derive from different origins and types of data (see section 3; only the magnetic basement top derives from a unique source). The variability in quality, periods of creation, owners, and compiler sensibility (human error) of the datasets produces large – not quantifiable – uncertainty affecting the input data. Furthermore, since most of the collected public data derive from interpretations of seismic profiles, the data in depth are strongly influenced by the velocity model used for the depth conversion. Unfortunately, most of the primary data are not provided along with the used velocity model and for this reason it was not possible to take into account this variable. Further, the study area is non-uniformly described by the dataset.

Each grid node is associated with one to four values of depth depending on the number of available identified sites (i.e., checkpoints from different sources) coinciding with that node. Hence, it turns out that it is possible to calculate the maximum and minimum depth of each surface at each node if at least two depth values are available in that specific node. . The range of depths at the nodes represents the uncertainty on the level description. Figure 12 shows the depth variation for the four surfaces calculated considering the maximum and minimum values for each node with at least two values . The bottoms of recent continental deposits (light blue band) and of Pleistocene unit (yellow band) show a small variation in depth (<2 km), with respect to the Pliocene (orange band - up to 6 km) and Carbonate (green band - up to 8 km) bases. Maps in Figure 13 illustrate

the geographic distribution of depth evaluation uncertainty. Interactive figures of each map reported in Figure 13 are available in the supplementary material (S1.zip).

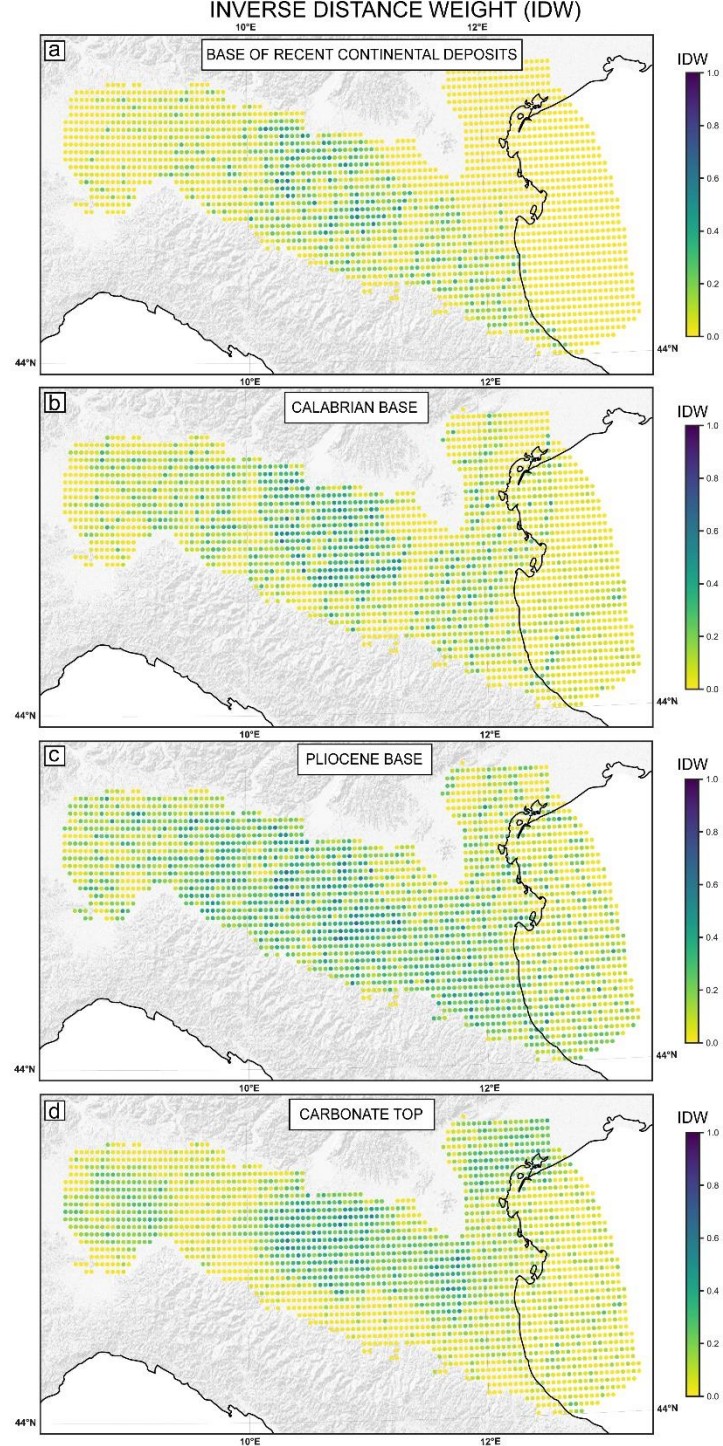

**Figure 11: The spatial distribution of the IDW for the four analysed surfaces The IDW values represent the confidence on the variable of interest (level depth) at each node of the grid (1 = max confidence).**

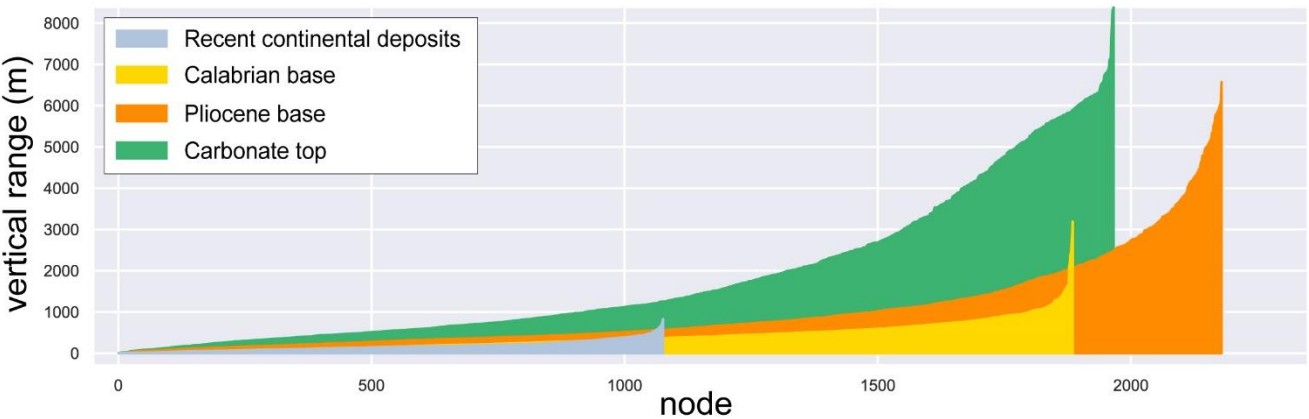

**Figure 12: Range of depths variation interpolated at each point of the evaluation grid based on the different datasets available for the four surfaces.**

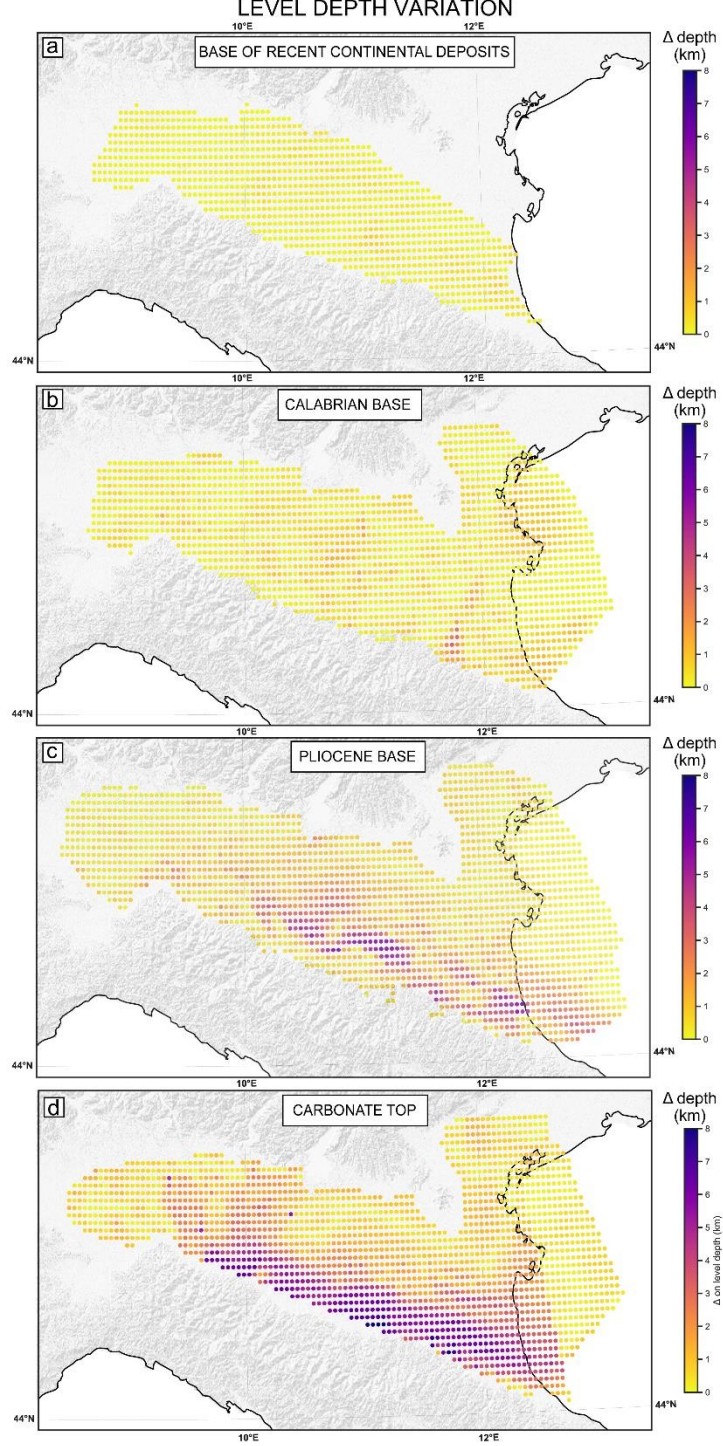

**Figure 13: Geographic distribution of the range of depths variation for the four analysed surfaces**

## 6 Derived data: methods and results

The construction of an accurate 3D geological model requires good coverage and consistency of the source data. Improving the multi-source data consistency helps to avoid errors and distortions during the 3D model generation. To ensure the most significant data coverage and consistency in the area, we revised, compared, and finally integrated the collected primitive data (i.e., wells, geological cross-sections, and maps).

Based on the data accuracy analysis and the uncertainty associated with level depth (see sections 5), we totally or partially removed the digitized primitive data showing the highest discrepancy with other data. For this purpose, as mentioned in the previous section, we assigned a priority order to the data according to their type. The highest priority is attributed to well data as they give the best depth constraint of the units (order 1 checkpoints). Hence, the well data represent the main control points of our model and none of them were excluded during the integration process. Regarding geological cross-sections and maps, if a difference in the depth of the surface is observed (Fig. 13), we kept those data showing the best fit with well data. In other cases, we kept the most recent data, except when they were very discordant from the average data (for example geological sections with a horizon depth very different from that indicated by other more reliable data) or from the predominant interpretative schemes (for example geological cross-sections with a tectonic style completely different from the preponderant interpretations). In particular, it should be pointed out that the base of recent continental deposits is defined from well data and the isobath map reported in Scardia et al. (2012) which integrates the aquifer data of the Lombardy (Regione Lombarida ENI-AGIP, 2002) and the Emilia Romagna Regions (Regione Emilia Romagna & ENI-AGIP, 1998). The inconsistent data removal was performed manually. These revised data integrated with the top of the magnetic basement, the only unit deriving from a single source (i.e., Cassano et al., 1986) and hence without any integration, were collected in ASCII format reporting the xyz coordinates of the digitized data.

We eventually constructed a 3D geological model of the rock volume interposed between the magnetic basement top and the topographic surface (Middle Triassic-present day) by means of the revised primitive data. To define the top of the modelled rock volume, since our study area is located both offshore and onshore (Fig. 2), we joined the land topography, deriving from a public digital elevation model of the whole Italian territory at an original resolution of 10 meters cell size (Tarquini et al., 2007), with the bathymetry of the offshore area (https://www.gebco.net/). The 3D geological model is made up of unfaulted surfaces obtained by gridding the revised and integrated multi-source primitive data. For the gridding process, we applied in Petrel® software a convergent interpolation algorithm with a 250 m sampling step. After gridding, we quality-checked and, where necessary, manually improved the modelled surfaces. The 3D model does not integrate any fault element. Thus, the modelled surfaces do not show any dislocation but sudden height differences in correspondence of the major tectonic structures, mostly where thrust sheets produce the vertical superimposition of the same unit (Fig. 14). The accuracy of the derived geological surfaces mainly depends on the quality and the quantity of source data (i.e., primitive data).

The aforementioned gridded surfaces represent the main boundaries that define units with different mechanical behaviour and subdivide the model into five sub-volumes, from top to bottom: the continental portion of the Quaternary deposits, the

Pleistocene Asti sands, the Pliocene sands (i.e., Santerno, Porto Corsini and Porto Garibaldi sandstones), the upper Eocene-to-Messinian siliciclastic formations (i.e., Gonfolite-Gallare Marls, Marnoso Arenacea Fm., San Donà Marls, Glauconie di

Cavanella Fm., Fusignano Fm., Sergnano gravels, etc.) and the carbonate-marly succession.

The base of the recent continental deposits (Fig. 14a) was reconstructed in a restricted area, where it was intercepted by wells and modelled in the literature (Scardia et al, 2012). Areas with recent prodelta, delta and marine sediments have been excluded (Fig. 14a). The base of the recent continental deposits consists of a slightly articulated surface that generally deepens from the peripheral sectors of the model area to the median areas of the Po valley. It is characterized by a depocenter area located

between the Emilia and Ferrara Arcs, where it reaches its maximum depth in the Parma area. The accuracy of this surface is generally intermediate with a strong constrain given by the isobath map digitized from Scardia et al. (2012) (Fig. 4).

The Calabrian base (Fig. 14b) generally delimits the lower contact between the Asti sands and the Gelasian and Pliocene sandstones, except where, due to the erosion affecting the anticline culminations (in correspondence with Emilia and Ferrara Arcs), the Pleistocene deposits are unconformably in contact with the Miocene ones (on the top of the Emilia Arc anticlines).

This surface progressively deepens from the model borders to the inner part of the Po Plain, except for the Ferrara Arc area, where some culminations can be observed. It reaches its maximum depth in the Parma-Mantova area, and in the coastal and northern Adriatic areas. The accuracy of this surface is variable, but generally well-constrained by most of the available geological cross-sections and numerous wells. Moreover, in the Brescia-Mantova area, it is constrained by an accurate depth contour map (ISPRA, 2015; Fig. 4c).

The Pliocene base covers almost the entire area except for the Emilia and Ferrara anticlines where it was eroded due to the tectonic uplift (Fig. 14c). This lithological boundary shows a rather articulated morphology characterized by pronounced culmination (located on the main anticline axes of the Emilia and the Ferrara Arcs) and depocenter areas (between the Emilia and Ferrara Arcs, immediately south of the Ferrara Arc, and in the coastal area). The accuracy of this surface is variable but it is generally constrained by a large number of wells, geological cross-sections, and by two subsurface geological maps covering

the entire model area (Bigi et al.,1990a, 1990b) and the GEOMOL project study area (ISPRA, 2015; Fig. 4c).

The Carbonate succession top separates the siliciclastic formations (upper Eocene-Present) from the carbonate and marly ones (Triassic to middle-upper Eocene in age). This surface deepens from NE to SW (beneath the northern Apennine front), with a local and pronounced elevation at the Ferrara Arc, where carbonates were markedly uplifted by the Apennine orogenic process (Fig. 14d). The accuracy of this surface is variable. It is constrained by several geological cross-sections but, due to its

considerable depth, by few deep well data, mostly located on the anticline culminations. Three geological maps located in the eastern portion of the model (Casero et al., 1990; Nicolich et al., 2004) and in the central portion (D'Ambrogi et al., 2023) constrain the surface of the carbonate succession top (Fig. 4c).

As mentioned above, the gridded magnetic basement top (Fig. 14e) is based on the published depth contour map by Cassano et al. (1986), consequently, its accurateness depends only on the source data accuracy.


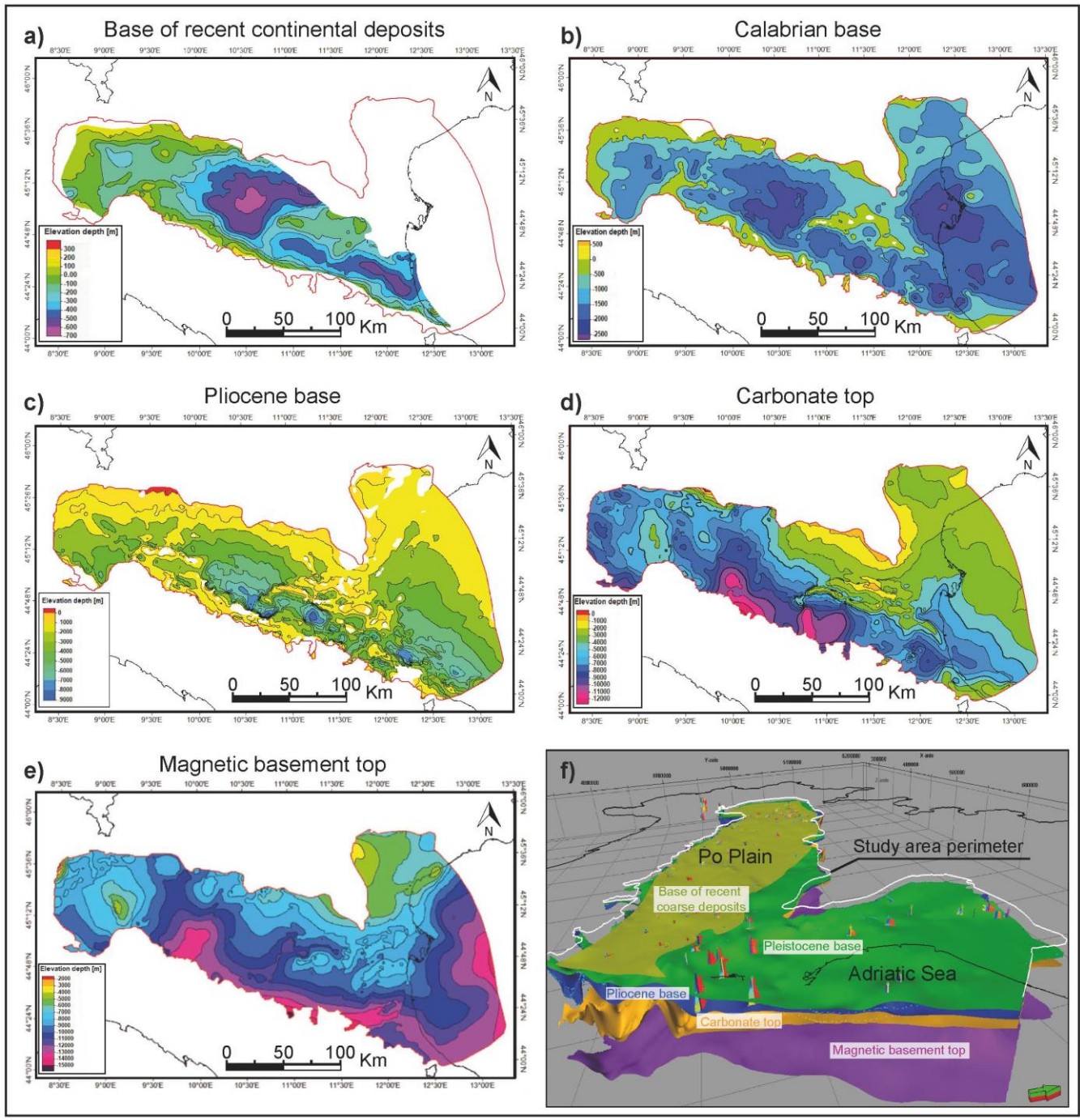

**Figure 14: a-e) Geological maps representing the gridded surfaces generated with the revised primitive data: a) Base of the recent continental deposits; b) Calabrian base; c) Pliocene base; d) Carbonate succession top; e) Magnetic basement top. f) Example of the data digitized and the geological model in 3D view cut along a SW-NE oriented section.**


**7 Data availability**

Primitive and derived data are available for open access in ASCII format at the following link: doi.org/10.5281/zenodo.8126519 (Livani et al., 2023). The format and the organization of data are explained in the related description file.

The dataset can be imported into any software handling 3D geological data, well data information and spatial vector data formats.

**8 Conclusions**

The database provides a collection of geological-geophysical data of the Po Plain subsurface and of the northern Adriatic Sea collected and digitized from the literature and from open repositories. We digitized data from the composite logs of 160 wells

drilled in the area. Borehole information (i.e., wellhead coordinates, rotary table elevation, measured depth, true depth, total depth and deviation survey) and Spontaneous Potential, Gamma Ray, and Sonic logs are provided in ASCII format. Five horizons were digitized from 61 geological cross-sections and from 10 geological maps, from the oldest to the youngest: the top of the magnetic basement, the top of the carbonate succession, the base of the Pliocene, the base of the early Pleistocene (i.e., near top Gelasian; see section 3) and the base of recent continental deposits. The digitized data are provided in ASCII

format reporting the xyz coordinates of the digitized surfaces.

Through an accuracy analysis performed on the primitive data and their subsequent processing, a new set of data has been created (i.e., derived data). Since the primary data show a depth uncertainty, we accurately revised the primary data by integrating only the data showing the best fitting. From these data, we generated a simplified 3D geological model characterized by several gridded surfaces of the main geological units (Fig. 14f).

Through the elaboration of the digitized logs is possible to directly extract geophysical and mechanical properties of the rock volume interposed between the gridded surfaces (e.g., P-wave velocity) and obtain further derived. For example, sonic velocities can be converted to mechanical rock properties, such as Young's modulus or Poisson ratio, that find applicability in geomechanical simulations which are performed to evaluate the ground subsidence/uplift phenomena (Carminati and Di Donato, 1999; Benetatos et al., 2017; Antoncecchi et al., 2021) or the change of stress field in a specific area in response to

natural processes or anthropic activities (e.g., Schutjens et al., 2010; Radwan and Sen, 2021; Hemami et al., 2021; Sangnimnuan et al., 2021).

The dataset developed in the present work supports application in a wide range of research areas with benefits for scientists, practitioners, and decision-makers. As an example, once populated with the values of seismic velocity, the 3D geological model can find several applications in seismological studies. It can be used to improve the procedure and reduce the

uncertainties during earthquake location, contribute to the calculation of more accurate focal mechanisms and perform wave-propagation and ground-motion simulations (e.g., Magistrale et al., 1996; Süss et al., 2001; Molinari et al., 2015; Livani et al., 2022). The 3D model also represents a starting model in perturbation studies, such as linearized inversions of travel times for

crustal velocities (e.g., Magistrale and Day, 1999) or for studies related to the seismic waveforms for crustal structure and, moreover, it can be used to derive densities and compare them to gravity observations (Roy and Clayton, 1999).

In conclusion, this database will be useful to better define and mitigate the possible natural and anthropogenic risks to preserve the environment and safeguard the social and economic interests of the territory contributing to a better and more efficient management of subsoil resources.

**Author contribution**

Michele Livani: data curation, formal analysis, investigation, methodology, validation, visualization, writing – original draft preparation; Lorenzo Petracchini: conceptualization, data curation, methodology, project administration, supervision, validation, visualization, writing – original draft preparation; Christoforos Benetatos: conceptualization, data curation, formal analysis, investigation, methodology, supervision, validation, visualization, writing – original draft preparation; Francesco Marzano: data curation, formal analysis, methodology, investigation, writing – review & editing; Andrea Billi:
conceptualization, project administration, supervision, methodology, writing – review & editing; Eugenio Carminati: conceptualization, project administration, supervision, writing – review & editing; Carlo Doglioni: project administration, supervision, writing – review & editing; Patrizio Petricca: data curation, formal analysis, validation, visualization, writing – original draft preparation; Roberta Maffucci: data curation, formal analysis, investigation, validation, writing – review & editing; Giulia Codegone: data curation, formal analysis, investigation, writing – review & editing; Vera Rocca:
conceptualization, supervision, writing – review & editing; Francesca Verga: conceptualization, project administration, supervision, writing – review & editing; Ilaria Antoncecchi: supervision, writing – review & editing

**Declaration of competing interest**

The authors declare that they have no known competing interests.


**Acknowledgments**

The authors acknowledge the General Direction for Infrastructure and Safety of the Ministry for Ecologic Transition (formerly Italian Economic Development Ministry) for their support to this research.

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
