# Peer review of "Subsurface geological and geophysical data from the Po Plain and the northern Adriatic Sea (north Italy)"

_Earth System Science Data, 2023_

## Author Comment (AC3)

Comments and Responses
Reviewer 1

**GENERAL COMMENTS**

**1.**

*Rev. 1: A general question is relative to the study area extent and to the choice of the available dataset. The authors include in their reconstructions and analysis only a slight part of the Venetian Friulian basin (east of the Schio-Vicenza fault). There are available data for at least three of their five analyzed horizons that cover the entire extent of this basin. Toscani et al., 2016 (Ital. J. Geosci., Vol. 135, No. 3), provide both isobath maps and geological cross sections of this basin. Also the cited paper of Nicolich et al., 2004 provide nice and complete data about some of the considered horizons in the Venetian basin subsoil. Moreover, the paper by Pola et al., 2014 (Ital. J. Geosci. (Boll. Soc. Geol. It.), Vol. 133, No. 2) provides a set of geological cross sections derived from seismic reflection profiles and well data that could allow to connect the surfaces through the Schio-Vicenza lineament (that apparently shouldn't be an "obstacle" for the interpolation process and reconstruction, given that it is a subvertical mainly strike slip fault and the authors declare that in their reconstructions fault displacements are treated as vertical steps).*

Authors: We thank the reviewer for the appropriate comment. Although we substantially agree with the reviewer, the study area has been selected primarily considering the well distribution. As explained within the manuscript, well data give better constraints with respect to depths derived by geological cross-sections or interpolated maps. During the 3D model construction, we used well data as main control points and for this reason, in the accuracy analysis, we assigned the higher specific weight to well data (i.e., first-order checkpoints) with respect to geological sections and maps (second- and third-order, respectively). As well evidenced in Figure 4, the well distribution is not homogenous and in the Venetian Friulian basin only one public well is present. Furthermore, the 3D model has been constructed using data from 5 lithological boundaries. Having an area with a subset of data surfaces will generate a non-coherent construction of the 3D model with surfaces (those without data in the Venetian Friulian basin) arbitrarily interpolated. In any case, the dataset will be implemented in the future and new data could certainly be integrated in the new version of the dataset. Concerning the data from Pola et al., 2014, we thank the reviewer for the useful information. We implemented our database with the geological sections provided in the manuscript. New tables 1-2, figure 4 and the data accuracy analysis have been now updated with these data. Please see changes within the text and the new references list.

**2.**

***Rev. 1:*** **The same happens for the subsurface maps provided by Amadori et al., 2019; also in this paper 2 of the 5 analyzed horizons are considered and coherently reconstructed throughout a wide area of the Po Plain, also covering the area of the GEOMOL project.**

Authors: Amadori et al., 2019 coherently reconstruct six unconformity-bounded sequences generating several isobath and isochore maps within the Plio-Pleistocene sedimentary succession. Considering the horizons that we defined to construct the proposed 3D model, only the *PL1u* (i.e., Pliocene Base) and potentially the *PS2u* unconformities (considering the *PS2u* equal to our Calabrian Base according the scheme adopted in our work, see lines 219-220) in Amadori et al. (2019) would be useful data for our dataset. Unfortunately, the isobath maps reported in Amadori et al. (2019), see Fig. 7 and Fig. E in the original paper, do not allow an accurate extrapolation of these information since it is not possible to define the depth reported in the maps. This is one of the reasons why some useful data, as indeed are those reported in Amadori et al. (2019), have been excluded in the database construction (see comment 5 to reviewer 1).

The Pliocene and Calabrian bases are two surfaces characterized by relatively good accuracy and with a small variation in depth (see Data accuracy paragraph). Hence, although our 3D model would have been certainly benefited from the Amadori et al. (2019) work, we consider the Pliocene and Calabrian bases sufficiently constrained.

**3.**

***Rev. 1:*** **Lastly, the authors do not consider and do not cite the outputs of the "Hotlime project" (see https://geoera.eu/projects/hotlime6/). In this case a depth converted surface of the top of the carbonates is provided almost all over the studied area but, surprisingly, it these relatively recent and well constrained data are not considered.**

Authors: We thank the reviewer for this very useful comment. The results of the "Hotlime project" has been recently integrated in the GO-PEG project (D'Ambrogi et al., 2023) and we implemented this data within our dataset. Figure 4, the new table 1 and the data accuracy analysis have been now updated with these data. Please see line 113, 141, 427, 629 and the new references list.

**4.**

***Rev. 1: In my opinion, the authors should better explain and clarify their choice and clarify it in the text by correcting or supplementing their statement (lines 108-109) "In this work, we refined***

*the previous regional 3D geological models extending the literature used to define the geometry of the geological surfaces".*

Authors: Done. We properly modified and integrated the sentence. Please see lines 117-122.

**5.**

*Rev. 1:* **Another comment/request of clarification deals with the data selection and their preliminary analysis. The authors declare that they graphically rearranged the sections, imperfections were reduced, parts of the sections were re-positioned and (line 364) they declare that some sections have been excluded in case of inconsistencies with the contiguous ones. In my opinion, the criteria at the base of this selection should be clarified or explained providing a couple of examples, because in this way it is not clear which section(s) they decide is "wrong" and if there's a quantitative parameter to define them as wrong (different depths of the horizons? Of all horizons or on one only? Which is the inconsistency amount to exclude a section? May be that on different sections there are big inconsistences on the carbonate top, but the Pleistocene base are comparable. How are treated the sections in this case? And which of the two is excluded?**

**I think that with a couple of examples, the reader could easily understand this point that quite heavily affects the dataset.**

Authors: We thank the reviewer for this comment. We modified the text according to the reviewer's comment adding some more information and some examples. Please see lines 410-418.

**6.**

*Rev. 1:* **A request of clarification is relative to the assumptions that are always connected with data choice and selection. All the subsurface maps and cross sections considered in this paper are depth converted yet. The depth conversion process of seismic reflection profiles is based on a velocity model that strongly affects the final result. In the framework of a so detailed and quantitative data analysis, it sounds a bit strange to me that the authors do not cite and do not consider at all the possible inconsistencies, differences and may be errors given by the native velocity models on whom sections and maps are based. The authors could consider to add a brief paragraph (or a table in the supplementary material) indicating the range of the velocity values (if and when available) used in the original papers from where they selected sections and maps.**

Authors: We thank the reviewer for this comment, and we agree with his/her point of view. We carefully check the data collected to get some information on the velocity model used to depth convert

the seismic profiles. Unfortunately, for almost all the public data we collected, no information on the velocity models used are provided. In accordance with the reviewer's comment, we have highlighted this aspect within the text. Please see line 533-536.

**7.**

***Rev. 1:*** **The last point deals with the chronostratigraphic subdivision (lines 189-200 in the manuscript). What the authors declare (pre- and post-2009 age of quaternary sediments) is true and can cause misunderstandings, but I think that maintaining a pre-2009 subdivision in a paper published in 2023 will cause more misunderstandings and is formally wrong. In this case I warmly suggest to the authors to do an effort and to use the current stratigraphic subdivision.**

Authors: We thank the reviewer for this comment. We modified the text according to the reviewer's comment using the term Calabrian Base instead of Pleistocene Base (similarly to what has been presented in Toscani et al., 2016) specifying that this horizon is generally represented by the Base of the Sabbie D'Asti. Please see lines 217-219 and elsewhere within the text and in the figures.

**MINOR/SPECIFIC POINTS**

**8.**

***Rev. 1: LINE 28: in my opinion "strategies to ensure the safety of urbanized areas and human activities" usually need much more detailed studies at a different scale. I would say that this study can be of help in defining and highlighting areas where data collection and more detailed studies are needed for…***

Authors: We have considered the reviewer's comment and we have modified the text. Indeed, specific and detailed studies should be carried out to properly set actions to reduce natural and anthropic risks. Nonetheless, in our opinion some of the data collected, as explained in the Conclusion paragraph, could be already useful to define regional first order strategies. Please see lines 28-30.

**9.**

***Rev. 1: LINE 47: add Cassano et al to the references cited***

Authors: Done. Please see line 49.

**10.**

***Rev. 1: LINE 87: same comment of line 28***

Authors: Done. Please see line 89-90.

**11.**

*Rev. 1: LINE 128: probably Turrini et al., 2016 could be added to the references cited*

Authors: Done. Please see line 133.

**12.**

*Rev. 1: LINE 130: in my opinion the best sections showing the outermost buried fronts of the two thrust belts are from Fantoni and Franciosi 2010 and Toscani et al., 2014*

Authors: Done. Please see line 145.

**13.**

*Rev. 1: LINE 129-132: if you change the order of the two sentences the pictures (2a and 2b) will be cited in the right order (2a before 2b) without any change in the general meaning of the paragraph.*

Authors: Done. Please see lines 142-145.

**14.**

*Rev. 1: LINE 140-143: a clarification of this paragraph would be of help. Probably the upper cycle is fed by the Apennine chain (as stated by the authors) close to the outcropping Northern Apennines, but throughout the Plio-Plesitocene the Alps are still providing sediments. In figs. 18, 20 and 21 of Ghielmi et al., 2013 all the entry points of the main sand systems are located close to the southern alps.*

Authors: We thank the reviewer for this comment and apologize for this oversight. The sentence has been modified. Please see lines 155-158.

**15.**

*Rev. 1: LINE 150: in the reference list, and in general in the paper, it should be added Fantoni et al., 2004 (Boll. Soc. Geol. It., 123 (2004), 463-476) where, in my opinion, one of the best stratigraphic sketch of the Southern Alps is presented.*

Authors: Done. Please see line 170.

**16.**

*Rev. 1: LINE 174: I think other papers dealing with recent tectonic activity of the Po Plain should be cited (Livio et al., 2009; Zuffeti and Bersezio, 2020; Bresciani and Perotti, 2014).*

Authors: Done. Please see line 192.

**17.**

*Rev. 1: LINE 209-211: I think this part should be moved at the beginning of the paper in order to declare since the beginning the possible lack of data*

Authors: We thank the reviewer for this useful comment. We modified the text according to the reviewer's comment. Please see line 106-109.

**18.**

*Rev. 1: TABLE 2 AND FIG4A: it would be useful to highlight which wells drills the 4 horizons considered in the study. Different colors (and symbols in the table) for wells drilling the carbonates, for wells drilling the base of the Pliocene etc could be provided*

Authors: We thank the reviewer for this useful comment. We modified Table 3 (the former table 2) according to the reviewer's comment with a new field showing the deepest unit encountered in wells. Furthermore, the new table 1 shows the list and type of data collected for each lithological boundary.

**19.**

*Rev. 1: LINE 364: if I understand correctly, this means that you have a dataset of all collected sections but then on each of them you exclude the horizons that for different reasons are not usable. This means that the stratigraphic horizons have been reconstructed using a non-homogeneous dataset. Why not providing four maps reporting ONLY the sections used for a given horizon?*

Authors: We thank the reviewer for the comment, and we hope that with the changes reported in comment 5 this misunderstanding related to primitive data is fixed. Anyhow, we are available for further clarification. In the primitive data we have excluded some data that show relevant incongruencies (for example vintage geological profiles with a dated interpretation of the subsoil). The new Table 1 shows the primitive data collected for each horizon.

**20.**

*Rev. 1: LINE 371-372: why not considering the surfaces from Amadori et al., 2019?*

Authors: Please see response in comment 2.

**21.**

*Rev. 1: LINE 377: even if you provide a detailed analysis of data accuracy, it would be useful to provide some examples and values of this check. For example taking a well far from the digitized sections and not included in the interpolation process and check at which vertical distance the reconstructed horizon is from the values indicated in the logs. Possibly taking a well placed on the top of an anticline and another one along the monocline.*

Authors: We thank the reviewer for this comment. In the Primitive data paragraph (i.e., Paragraph 4), we made a non-analytical check of the geological map useful only to verify their consistencies with other data. In our opinion, the data accuracy analysis shows well the discrepancies and the depth variation within the collected data. Furthermore, in the supplementary material we provided (S1.zip file) detailed statistical information for each node of the interpolation grid.

**22.**

*Rev. 1: FIG9: just a suggestion (after a trial, the authors will decide if to follow this suggestion or not). Why not plotting on this map the section traces and the map polygons? If graphically possible, this would help in providing the readers an immediate view of the correspondence between digitized sections and maps and the reconstructed surfaces.*

Authors: We thank the reviewer for this comment. We tried to modify the figure but, with the changes suggested, the data (we necessarily have to put even the wells since they represent the main control points) would make the figure less readable. Furthermore, we remind that the data accuracy has been carried out on the primary data and not on the derived data, and figure 4 and the new Table 1 show the data collected. Indeed, the data accuracy has been one important toll for us to select the best data to derive the 3D geological model.

**23.**

*Rev. 1: Paragraph 5.3 LINES 472-483: this part and its significance should be reorganized/rewritten and better clarified. I agree that if you have two (or more) vertical values in correspondence of a node, you can calculate the maximum and minimum depth of the horizon in correspondence of the node, but it is not that clear why the range of depths at the nodeS (you mean all nodes?) quantifies the uncertainty of the level. I think that the uncertainty is associated with a single point. On a given level you could have nodes with different values along the vertical, and nodes where the vertical values are coincident. This means, in my opinion, that in the first point the uncertainty is high, while in the second is low. The level of uncertainty is different along the level.*

Authors: We thank the reviewer for this comment. Indeed, Figure 13 shows exactly the reviewer's opinion. Along each horizon (surface), the uncertainty in depth is associated to each node and it varies from 0 to 8 km. Hence, for the same horizon, where 2 or more vertical values are available, it is possible to have node with 0 km or >0 km of vertical variation in case data are coincident or not, respectively. We have properly modified the text. We rephrased Paragraph 5.3.

**24.**

*Rev. 1: Secondly, I would provide a percentage of these variations with respect to the maximum and minimum depth. The small (or big) variations in depth of different horizons should be evaluated with respect to the depth of the horizon. 2 km of variations on an horizon that reaches a maximum depth of 2 km is a 100% of variation; the same value (2km) if referred to a very deep horizon could be a 20% of variation so indicating a better coherence between different guesses*

Authors: We thank the reviewer for this comment. The proposed suggestion is indeed interesting but since the depth variation of the single horizon (in particular the Pliocene base and the top of the carbonate succession) is relevant (for instance, the Carbonate top ranges roughly from 12 km to 0 km), a percentage variation of each node with respect to the maximum and minimum depth would be, in our opinion, misleading (e.g., the single node might be at a depth very different from the maximum and minimum values of the entire horizon). The same problem will arise if we consider the average value of the depth. Any further suggestions aimed at improving the manuscript quality will be certainly taken into consideration.

---

## Author Comment (AC4)

**GENERAL COMMENTS**

**1.**

*Rev. 2: I understand the difficulty of resuming the regional geology, but there are several imprecisions that may have a strong impact on the definition of the mapped surfaces. For example, where it is incorrectly written that the carbonate substratum is Triassic to middle Eocene, you are neglecting the early phase of Alpine Orogeny (Aptian-Maastrichtian), which in turn provides one of the most evident stratigraphic and seismic horizons in the Po Plain subsurface (top of Maiolica Fm). Actually, apart few isolated carbonate platforms in the distal foreland (e.g., Bagnolo platform), since the Aptian the Po Plain Alpine foredeep sedimentation is mostly characterized by marls (e.g., Di Giulio et al., 2001; Tremolada et al., 2008; Sciunnach et al., 2023). What you mean as Carbonate succession top is actually the top of Scaglia Fm, (Scaglia is marl to claystone sediment) and it marks the onset of full Alpine collision (e.g., a tectono-stratigraphic sequence boundary).*

Authors: We thank the reviewer for the comment. As explained within the manuscript, we have used public data to build our dataset and the 3D geological model. We generally agree with the reviewer but, in the literature, data (maps and geological cross sections) report generally the top of the Cretaceous (very rarely the Top of the Maiolica Fm.) or the top of the Scaglia Fm. as the end of the carbonate succession. Hence, we were forced to use these data in order to get a good coverage of this important horizon. Indeed, the Scaglia Fm. is characterized by a certain amount of clay, mainly in its Cenozoic phase, but in our work, we were interested in defining those surfaces representing important boundaries with respect to the mechanical properties of the geological succession (indeed the Scaglia Fm. shows this properties). The new Figure S.2 shows well the variation of the sonic log (and hence the mechanical properties) along the horizons selected for our database. In conclusion, our top of the carbonate succession horizon represents a mechanical boundary dividing a mainly carbonate succession (below) and a mainly siliciclastic succession (above) even not always defined by the Scaglia Fm. (in some well data, older carbonate formation where directly covered by siliciclastic units). We have now better explained this procedure following the reviewer's suggestion (e.g., see lines 315-317, 372-377, 402-403 596).

**2.**

*Rev. 2: It is understandable, as correctly written in the text, the need of focussing on the main previous studies, selected for extension and detail, however I've missed the pioneering studies on*

*the Po Plain shallow aquifers (ENI-RER, 1998; ENI-RL, 2002; Irace et al., 2010), which had a great importance in the definition of the most recent surfaces, whose one is possibly your base of the coarse-grained alluvial deposits. Various papers have also been published about this surface (R-surface, e.g., Muttoni et al., 2003; Garzanti et al., 2011; Scardia et al., 2012; Ghielmi et al., 2013).*

Authors: We thank the reviewer for pointing out these literature works that are valuable for our model. The new base of the recent continental deposits was implemented and integrated with the isobath map by Scardia et al. (2012) which integrates data from the Emilia Romagna (Regione Emilia Romagna & ENI-AGIP, 1998) and Lombardy Regions (Regione Lombardia & ENI-AGIP, 2002). We have chosen this map over other recent works as it broadly covers the study area, it is highly concordant with well data, and it represents the base of the recent continental deposits. Please see line 424 and elsewhere within the text and in the figures.

**3.**

*Rev. 2: About the surfaces you chose to focus on, I've found the stratigraphic approach not accurate: you define them as geological or lithostratigraphic surfaces, the former term being imprecise and the latter incorrect or not clear. Are they operational surfaces adequate for practical applications or have they been chosen because are more easily recognizable in well and geophysical data? I believe in the second hypothesis, but on the whole the stratigraphic meaning of these surfaces should be addressed more clearly and your choice (lithostratigraphic vs other kind) should be discussed according to the expected application of your dataset.*

Authors: We thank the reviewer for the comment. We have properly modified the text following the reviewer's suggestion. Please see lines 118-122, 315-317, 5372-376, 595-596, 400, and comment 1 to reviewer 2.

**4.**

*Rev. 2: Specifically, the recognized surfaces can be considered as sequence boundaries with chronostratigraphic value. In this sense, the reference to Ricci Lucchi (1986) is misleading and obsolete, because it is based on lithostratigraphic units established for ENI operational purposes, and in my opinion not very useful in defining stratigraphic surfaces with chronostratigraphic value or regional extension. It would be more useful to rely on the more recent syntheses of Fantoni and Ghielmi, which frame the old ENI lithostratigraphic units into the modern sequence stratigraphy. For example, the top of the carbonates, even if associated to the top of Maiolica Fm (see my previous comment) or to the top of the Scaglia, can be considered a tectono-stratigraphic sequence*

*boundary produced by the onset of an Alpine tectonic phase (eoAlpine or neoAlpine, respectively) and the associated pulses of terrigenous input into the basin. Also the base of Pliocene is a tectono-stratigraphic surface (e.g., Ghielmi et al., 2013), which documents the effect of Apennine tectonics on the Po Plain, modifying the geometry of the previous basin.*

*The most recent surface, which deserves a more detailed discussion, is a climate sequence boundary (R-surface), well defined in the works of the Po Plain aquifers and in the already mentioned following scientific articles.*

Authors: We thank the reviewer for the comment. We have better explained the meaning of our subdivision and properly modified the text following the reviewer's suggestion. Please see comments 1-3 to Reviewer 2.

**SPECIFIC POINTS**

**5.**

*Rev. 2: The magnetic basement is called Variscan (line 307) or Hercinian (lines 148, 152; Fig. 2): the former is preferable and should be used consistently in the text.*

Authors: We thank the reviewer for the comment. We have properly modified the text following the reviewer's suggestion. Please see lines 165, 170, and 337.

**6.**

*Rev. 2: For the alluvial deposits you mention as youngest surface the correct term would be "coarse-grained" (not just coarse) and, if we use a sequence stratigraphy approach (R-surface of Muttoni et al., 2003), at east of Ferrara they pass to shallow-water fine-grained marine sediments (Scardia et al., 2012; Ghielmi et al., 2013;) and at Venice they are prodelta turbidites (Muttoni et al., 2018). Lithostratigraphically they would be different units but apparently you are putting them altogether into a same sequence. This is another example about the importance of defining and justifying the stratigraphic approach you want to use and applying it correctly.*

Authors: We thank the reviewer for the comment. In our work, we have reconstructed the base of the recent continental deposits. The new surface has been limited in a restricted area, where the base of recent continental deposits was intercepted by wells and/or other public data (Scardia et al., 2012; Regione Emilia Romagna & ENI-AGIP, 1998; Regione Lombardia & ENI-AGIP, 2002. Please see lines 578-581, 600-603, and new Figure 14a

**7.**

*Rev. 2: Line 104: "geological maps". Perhaps you mean stratigraphic maps or isobath maps, or better use just maps.*

Authors: We properly modified the text. Please see line 109.

**8.**

*Rev. 2: Line 146: carbonate substratum. See my comments about Maiolica and Scaglia.*

Authors: Please see author's answer to comment 1 of Reviewer 2. We properly modified the text according to the reviewer's comment (line 164).

**9.**

*Rev. 2: Line 196: what do you mean with the reference to ENI S.P.A.? It is not in the reference list and it is not clear. Anyway, I agree with your discussion about the formal base of the Quaternary period, but the point here is how ENI defined the base of Quaternary in its wells. As far as I know, ENI's base of Quaternary was defined by the FO of Hyalinea balthica in the Mediterranean Sea following the recommendations of the 18th International Geological Congress (London, 1948; see for example discussion in Gradstein et al., 2012, p. 980). By the way, the FO of H. Balthica in the Po basin has been recently dated at ca. 1.9 Ma (Monesi et al., 2018) in the Arda River series. On the whole, you should discuss what you use actually as base of Quaternary also according to the stratigraphic data you're including in your dataset.*

Authors: We thank the reviewer for the comment. We modified the text according to the reviewer's comment using the term Calabrian Base instead of Pleistocene Base (similarly to what has been presented in Toscani et al., 2016). Please see lines 217-219 and elsewhere within the text and in the figures.

**10.**

*Rev. 2: Line 213: geologic maps, see my previous comment*

Authors: We properly modified the text. Please see line 232.

**11.**

*Rev. 2: Line 547: the typical Carbonate succession top in the Po basin is the top of Maiolica Fm (e.g., Fantoni and Franciosi (2010). The one you are using is the top of Scaglia Rossa. Actually, the Gallare Marls are also known as Scaglia Cinerea/Variegata on correlated exposures in the Alps and Apennines.*

Authors: Please see author's answer to comment 1 of Reviewer 2. We properly modified the text according to the reviewer's comment (please see lines 315-317, 372-377, 402-403 596).